# Improved Sample Complexities for Deep Networks and Robust Classification via an All-Layer Margin

**Colin Wei**
Stanford University
colinwei@stanford.edu

**Tengyu Ma**
Stanford University
tengyuma@stanford.edu

## Abstract

For linear classifiers, the relationship between (normalized) output margin and generalization is captured in a clear and simple bound – a large output margin implies good generalization. Unfortunately, for deep models, this relationship is less clear: existing analyses of the output margin give complicated bounds which sometimes depend exponentially on depth. In this work, we propose to instead analyze a new notion of margin, which we call the "all-layer margin." Our analysis reveals that the all-layer margin has a clear and direct relationship with generalization for deep models. This enables the following concrete applications of the all-layer margin: 1) by analyzing the all-layer margin, we obtain tighter generalization bounds for neural nets which depend on Jacobian and hidden layer norms and remove the exponential dependency on depth 2) our neural net results easily translate to the adversarially robust setting, giving the first direct analysis of robust test error for deep networks, and 3) we present a theoretically inspired training algorithm for increasing the all-layer margin. Our algorithm improves both clean and adversarially robust test performance over strong baselines in practice.

## 1 Introduction

The most popular classification objectives for deep learning, such as cross entropy loss, encourage a larger output margin – the gap between predictions on the true label and and next most confident label. These objectives have been popular long before deep learning was prevalent, and there is a long line of work showing they enjoy strong statistical guarantees for linear and kernel methods (Bartlett and Mendelson, 2002; Koltchinskii et al., 2002; Hofmann et al., 2008; Kakade et al., 2009). These guarantees have been used to explain the successes of popular algorithms such as SVM (Boser et al., 1992; Cortes and Vapnik, 1995).

For linear classifiers, the relationship between output margin and generalization is simple and direct – generalization error is controlled by the output margins *normalized by the classifier norm*. Concretely, suppose we have $n$ training data points each with norm 1, and let $\gamma_i$ be the output margin on the $i$-th example. With high probability, if the classifier perfectly fits the training data, we obtain[1]

$$\text{Test classification error} \lesssim \frac{1}{n}\sqrt{\sum_{i=1}^{n}\left(\frac{\text{classifier norm}}{\gamma_i}\right)^2} + \text{low order terms} \qquad (1.1)$$

For deeper models, the relationship between output margin and generalization is unfortunately less clear and interpretable. Known bounds for deep nets normalize the output margin by a quantity that either scales exponentially in depth or depends on complex properties of the network (Neyshabur et al., 2015; Bartlett et al., 2017; Neyshabur et al., 2017b; Golowich et al., 2017; Nagarajan and Kolter, 2019; Wei and Ma, 2019). This is evidently more complicated than the linear case in (1.1). These complications arise because for deep nets, it is unclear how to properly normalize the output

---

[1]This is a stronger version of the classical textbook bound which involves the min margin on the training examples. We present this stronger version because it motivates our work better. It can be derived from the results of Srebro et al. (2010).

margin. In this work, we remedy this issue by proposing a new notion of margin, called "all-layer margin", which we use to obtain simple guarantees like (1.1) for deep models. Let $m_i$ be the all-layer margin for the $i$-th example. We can simply normalize it by the sum of the complexities of the weights (often measured by the norms or the covering number) and obtain a bound of the following form:

$$\text{Test error} \lesssim \frac{1}{n} \sqrt{\sum_{i=1}^{n} \left( \frac{\text{sum of the complexities of each layer}}{m_i} \right)^2} + \text{low order terms} \qquad (1.2)$$

As the name suggests, the all-layer margin considers all layers of the network simultaneously, unlike the output margin which only considers the last layer. We note that the *definition* of the all-layer margin is the key insight for deriving (1.2) – given the definition, the rest of the proof follows naturally with standard tools. (Please see equation (2.2) for a formal definition of the margin, and Theorem 2.1 for a formal version of bound (1.2).) To further highlight the good statistical properties of the all-layer margin, we present three of its concrete applications in this paper.

1. By relating all-layer margin to output margin and other quantities, we obtain improved generalization bounds for neural nets. In Section 3, we derive an analytic lower bound on the all-layer margin for neural nets with smooth activations which depends on the output margin normalized by other data-dependent quantities. By substituting this lower bound into (1.2), we obtain a generalization bound in Theorem 3.1 which avoids the exponential depth dependency and has tighter data-dependent guarantees than (Nagarajan and Kolter, 2019; Wei and Ma, 2019) in several ways. First, their bounds use the same normalizing quantity for each example, whereas our bounds are tighter and more natural because we use a different normalizer for each training example – the local Lipschitzness for that particular example. Second, our bound depends on the empirical distribution of some complexity measure computed for each training example. When these complexities are small for each training example, we can obtain convergence rates faster than $1/\sqrt{n}$. We provide a more thorough comparison to prior work in Section 3.

Furthermore, for relu networks, we give a tighter generalization bound which removes the dependency on inverse pre-activations suffered by (Nagarajan and Kolter, 2019), which they showed to be large empirically (see Section B.1). The techniques of (Wei and Ma, 2019) could not remove this dependency because they relied on smooth activations.

2. We extend our tools to give generalization bounds for adversarially robust classification error which are analogous to our bounds in the standard setting. In Section 4, we provide a natural extension of our all-layer margin to adversarially robust classification. This allows us to translate our neural net generalization bound, Theorem 3.1, directly to adversarially robust classification (see Theorem 4.1). The resulting bound takes a very similar form as our generalization bound for clean accuracy – it simply replaces the data-dependent quantities in Theorem 3.1 with their worst-case values in the adversarial neighborhood of the training example. As a result, it also avoids explicit exponential dependencies on depth. As our bound is the first direct analysis of the robust test error, it presents a marked improvement over existing work which analyze loose relaxations of the adversarial error (Khim and Loh, 2018; Yin et al., 2018). Finally, our analysis of generalization for the clean setting translates directly to the adversarial setting with almost no additional steps. This is an additional advantage of our all-layer margin definition.

3. We design a training algorithm that encourages a larger all-layer margin and demonstrate that it improves empirical performance over strong baselines. In Section 5, we apply our regularizer to WideResNet models (Zagoruyko and Komodakis, 2016) trained on the CIFAR datasets and demonstrate improved generalization performance for both clean and adversarially robust classification. We hope that these promising empirical results can inspire researchers to develop new methods for optimizing the all-layer margin and related quantities.

## 1.1 ADDITIONAL RELATED WORK

Zhang et al. (2016); Neyshabur et al. (2017a) note that deep learning often exhibits statistical properties that are counterintuitive to conventional wisdom. This has prompted a variety of new perspectives for studying generalization in deep learning, such as implicit or algorithmic regularization (Gunasekar et al., 2017; Li et al., 2017; Soudry et al., 2018; Gunasekar et al., 2018), new analyses of interpolating classifiers (Belkin et al., 2018; Liang and Rakhlin, 2018; Hastie et al., 2019; Bartlett

et al., 2019), and the noise and stability of SGD (Hardt et al., 2015; Keskar et al., 2016; Chaudhari et al., 2016). In this work, we adopt a different perspective for analyzing generalization by studying a novel definition of margin for deep models which differs from the well-studied notion of output margin. We hope that our generalization bounds can inspire the design of new regularizers tailored towards deep learning. Classical results have bounded generalization error in terms of the model's output margin and the complexity of its prediction (Bartlett and Mendelson, 2002; Koltchinskii et al., 2002), but for deep models this complexity grows exponentially in depth (Neyshabur et al., 2015; Bartlett et al., 2017; Neyshabur et al., 2017b; Golowich et al., 2017). Long and Sedghi (2019) prove bounds for convolutional networks which scale with the number of parameters and their distance from initialization. Recently, Nagarajan and Kolter (2019); Wei and Ma (2019) derived complexity measures in terms of hidden layer and Jacobian norms which avoid the exponential dependence on depth, but their proofs require complicated techniques for controlling the complexity of the output margin. Neyshabur et al. (2017a); Arora et al. (2018) also provide complexity measures related to the data-dependent stability of the network, but the resulting bounds only apply to a randomized or compressed version of the original classifier. We provide a simple framework which derives such bounds for the original classifier. Novak et al. (2018); Javadi et al. (2019) study stability-related complexity measures empirically.

A recent line of work establishes rigorous equivalences between logistic loss and output margin maximization. Soudry et al. (2018); Ji and Telgarsky (2018) show that gradient descent implicitly maximizes the margin for linearly separable data, and Lyu and Li (2019) prove gradient descent converges to a stationary point of the max-margin formulation for deep homogeneous networks. Other works show global minimizers of regularized logistic loss are equivalent to margin maximizers, in linear cases (Rosset et al., 2004) and for deep networks (Wei et al., 2018; Nacson et al., 2019). A number of empirical works also suggest alternatives to the logistic loss which optimize variants of the output margin (Sun et al., 2014; Wen et al., 2016; Liu; Liang et al., 2017; Cao et al., 2019). The neural net margin at intermediate and input layers has also been studied. Elsayed et al. (2018) design an algorithm to maximize a notion of margin at intermediate layers of the network, and Jiang et al. (2018) demonstrate that the generalization gap of popular architectures can empirically be predicted using statistics of intermediate margin distributions. Verma et al. (2018) propose a regularization technique which they empirically show improves the structure of the decision boundary at intermediate layers. Sokolić et al. (2017) provide generalization bounds based on the input margin of the neural net, but these bounds depend exponentially on the dimension of the data manifold. These papers study margins defined for individual network layers, whereas our all-layer margin simultaneously considers all layers. This distinction is crucial for deriving our statistical guarantees.

A number of recent works provide negative results for adversarially robust generalization (Tsipras et al., 2018; Montasser et al., 2019; Yin et al., 2018; Raghunathan et al., 2019). We provide positive results stating that adversarial test accuracy can be good if the adversarial all-layer margin is large on the training data. Schmidt et al. (2018) demonstrate that more data may be required for generalization on adversarial inputs than on clean data. Montasser et al. (2019) provide impossiblity results for robust PAC learning with proper learning rules, even for finite VC dimension hypothesis classes. Zhang et al. (2019) consider the trade-off between the robust error and clean error. Farnia et al. (2018) analyze generalization for specific adversarial attacks and obtain bounds depending exponentially on depth. Yin et al. (2018); Khim and Loh (2018) give adversarially robust generalization bounds by upper bounding the robust loss via a transformed/relaxed loss function, and the bounds depend on the product of weight matrix norms. Yin et al. (2018) also show that the product of norms is inevitable if we go through the standard tools of Rademacher complexity and the output margin. Our adversarial all-layer margin circumvents this lower bound because it considers all layers of the network rather than just the output.

## 1.2 Notation

We use the notation $\{a_i\}_{i=1}^k$ to refer to a sequence of $k$ elements $a_i$ indexed by $i$. We will use $\circ$ to denote function composition: $f \circ g(x) = f(g(x))$. Now for function classes $\mathcal{F}, \mathcal{G}$, define $\mathcal{F} \circ \mathcal{G} \triangleq \{f \circ g : f \in \mathcal{F}, g \in \mathcal{G}\}$. We use $D_h$ to denote the partial derivative operator with respect to variable $h$, and thus for a function $f(h_1, h_2)$, we use $D_{h_i} f(h_1, h_2)$ to denote the partial derivative of $f$ with respect to $h_i$ evaluated at $(h_1, h_2)$. We will use $\| \cdot \|$ to denote some norm. For a function $f$ mapping between normed spaces $\mathcal{D}_I, \mathcal{D}_O$ with norms $\| \cdot \|_I, \| \cdot \|_O$, respectively,

define $\|f\|_{\text{op}} \triangleq \sup_{x \in \mathcal{D}_I} \frac{\|f(x)\|_O}{\|x\|_I}$, which generalizes the operator norm for linear operators. Let $\|M\|_{\text{fro}}, \|M\|_{1,1}$ denote the Frobenius norms and the sum of the absolute values of the entries of $M$, respectively. For some set $\mathcal{S}$ (often a class of functions), we let $\mathcal{N}_{\|\cdot\|}(\epsilon, \mathcal{S})$ be the covering number of $\mathcal{S}$ in the metric induced by norm $\|\cdot\|$ with resolution $\epsilon$. For a function class $\mathcal{F}$, let $\mathcal{N}_\infty(\epsilon, \mathcal{F})$ denote the covering number of $\mathcal{F}$ in the metric $d_\infty(f, \hat{f}) = \sup_x \|f(x) - \hat{f}(x)\|$. For a function $f$ and distribution $P$, we use the notation $\|f\|_{L_q(P)} \triangleq (\mathbb{E}_{x \sim P}[|f(x)|^q])^{1/q}$. We bound generalization for a test distribution $P$ given a set of $n$ training samples, $P_n \triangleq \{(x_i, y_i)\}_{i=1}^n$ where $x \in \mathcal{D}_0$ denotes inputs and $y \in [l]$ is an integer label. We will also use $P_n$ to denote the uniform distribution on these training samples. For a classifier $F : \mathcal{D}_0 \to \mathbb{R}^l$, we use the convention that $\max_{y' \in [l]} F(x)_{y'}$ is its predicted label on input $x$. Define the 0-1 prediction loss $\ell_{0\text{-}1}(F(x), y)$ to output 1 when $F$ incorrectly classifies $x$ and 0 otherwise.

## 2 WARMUP: SIMPLIFIED ALL-LAYER MARGIN AND ITS GUARANTEES

Popular loss functions for classification, such as logistic and hinge loss, attempt to increase the *output* margin of a classifier by penalizing predictions that are too close to the decision boundary. Formally, consider the multi-class classification setting with a classifier $F : \mathcal{D}_0 \to \mathbb{R}^l$, where $l$ denotes the number of labels. We define the output margin on example $(x, y)$ by $\gamma(F(x), y) \triangleq \max\{0, F(x)_y - \max_{y' \neq y} F(x)_{y'}\}$.

For shallow models such as linear and kernel methods, the output margin maximization objective enjoys good statistical guarantees (Kakade et al., 2009; Hofmann et al., 2008). For deep networks, the statistical properties of this objective are less clear: until recently, statistical guarantees depending on the output margin also suffered an exponential dependency on depth (Bartlett et al., 2017; Neyshabur et al., 2017b). Recent work removed these dependencies but require technically involved proofs and result in complicated bounds depending on numerous properties of the training data (Nagarajan and Kolter, 2019; Wei and Ma, 2019).

In this section, we introduce a new objective with better statistical guarantees for deep models (Theorem 2.1) and outline the steps for proving these guarantees. Our objective is based on maximizing a notion of margin which measures the stability of a classifier to *simultaneous* perturbations at all layers. Suppose that the classifier $F(x) = f_k \circ \cdots \circ f_1(x)$ is computed by composing $k$ functions $f_k, \ldots, f_1$, and let $\delta_k, \ldots, \delta_1$ denote perturbations intended to be applied at each hidden layer. We recursively define the perturbed network output $F(x, \delta_1, \ldots, \delta_k)$ by

$$\begin{aligned} h_1(x, \delta) &= f_1(x) + \delta_1 \|x\|_2 \\ h_i(x, \delta) &= f_i(h_{i-1}(x, \delta)) + \delta_i \|h_{i-1}(x, \delta)\|_2 \\ F(x, \delta) &= h_k(x, \delta) \end{aligned} \tag{2.1}$$

The all-layer margin will now be defined as the minimum norm of $\delta$ required to make the classifier misclassify the input. Formally, for classifier $F$, input $x$, and label $y$, we define

$$m_F(x, y) \triangleq \min_{\delta_1, \ldots, \delta_k} \sqrt{\sum_{i=1}^k \|\delta_i\|_2^2} \tag{2.2}$$
$$\text{subject to } \arg\max_{y'} F(x, \delta_1, \ldots, \delta_k)_{y'} \neq y$$

Note that the constraint that $F(x, \delta)$ misclassifies $x$ is equivalent to enforcing $\gamma(F(x), y) \leq 0$. Furthermore, $m_F$ is strictly positive if and only if the unperturbed prediction $F(x)$ is correct. Here multiplying $\delta_i$ by the previous layer norm $\|h_{i-1}(x, \delta)\|_2$ is important and intuitively balances the relative scale of the perturbations at each layer. We note that the definition above is simplified to convey the main intuition behind our results – to obtain the tightest possible bounds, in Sections 3 and 4, we use the slightly more general $m_F$ defined in Section A.

Prior works have studied, both empirically and theoretically, the margin of a network with respect to single perturbations at an intermediate or input layer (Sokolić et al., 2017; Novak et al., 2018; Elsayed et al., 2018; Jiang et al., 2018). Our all-layer margin is better tailored towards handling the compositionality of deep networks because it considers simultaneous perturbations to all layers, which is crucial for achieving its statistical guarantees.

Formally, let $\mathcal{F} \triangleq \{f_k \circ \cdots \circ f_1 : f_i \in \mathcal{F}_i\}$ be the class of compositions of functions from function classes $\mathcal{F}_1, \ldots, \mathcal{F}_k$. We bound the population classification error for $F \in \mathcal{F}$ based on the distribution of $m_F$ on the training data and the sum of the complexities of each layer, measured via covering numbers. For simplicity, we assume the covering number of each layer scales as $\log \mathcal{N}_{\|\cdot\|_{\mathrm{op}}}(\epsilon, \mathcal{F}_i) \le \lfloor \mathcal{C}_i^2/\epsilon^2 \rfloor$ for some complexity $\mathcal{C}_i$, which is common for many function classes.

**Theorem 2.1** (Simplified version of Theorem A.1). *In the above setting, with probability $1 - \delta$ over the draw of the training data, all classifiers $F \in \mathcal{F}$ which achieve training error 0 satisfy*

$$\mathbb{E}_P[\ell_{0\text{-}1}(F(x), y)] \lesssim \frac{\sum_i \mathcal{C}_i}{\sqrt{n}} \sqrt{\mathbb{E}_{(x,y) \sim P_n} \left[ \frac{1}{m_F(x,y)^2} \right]} \log^2 n + \zeta$$

*where $\zeta \triangleq O\left( \frac{\log(1/\delta) + \log n}{n} \right)$ is a low-order term.*

In other words, generalization is controlled by the sum of the complexities of the layers and the quadratic mean of $1/m_F$ on the training set. Theorem A.1 generalizes this statement to provide bounds which depend on the $q$-th moment of $1/m_F$ for any integer $q > 0$ and converge at rates faster than $1/\sqrt{n}$. For neural nets, $\mathcal{C}_i$ scales with weight matrix norms and $1/m_F$ can be upper bounded by a polynomial in the Jacobian and hidden layer norms and output margin, allowing us to avoid an exponential dependency on depth when these quantities are well-behaved on the training data.

We will break down the proof of Theorem 2.1 into two simple parts. The first part hinges on showing that $m_F$ has low complexity which scales with the sum of the complexities at each layer. The second part relates $m_F$ to the 0-1 loss using the simple fact that $m_F(x, y)$ is nonzero if and only if $F$ correctly classifies $x$.

**Lemma 2.1** (Complexity Decomposition Lemma). *Let $m \circ \mathcal{F} = \{m_F : F \in \mathcal{F}\}$ denote the family of all-layer margins of function compositions in $\mathcal{F}$. Then*

$$\log \mathcal{N}_\infty \left( \sqrt{\sum_i \epsilon_i^2}, m \circ F \right) \le \sum_i \log \mathcal{N}_{\|\cdot\|_{\mathrm{op}}}(\epsilon_i, \mathcal{F}_i) \tag{2.3}$$

*The covering number of an individual layer commonly scales as $\log \mathcal{N}_{\|\cdot\|_{\mathrm{op}}}(\epsilon_i, \mathcal{F}_i) \le \lfloor \mathcal{C}_i^2/\epsilon_i^2 \rfloor$. In this case, for all $\epsilon > 0$, we obtain $\log \mathcal{N}_\infty (\epsilon, m \circ \mathcal{F}) \le \left\lfloor \frac{(\sum_i \mathcal{C}_i)^2}{\epsilon^2} \right\rfloor$.*

Lemma 2.1 shows that the complexity of $m_F$ scales linearly in depth for any choice of layers $\mathcal{F}_i$. In sharp contrast, lower bounds show that the complexity of the output margin scales exponentially in depth via a product of Lipschitz constants of all the layers (Bartlett et al., 2017; Golowich et al., 2017). Our proof only relies on basic properties of $m_F$, indicating that $m_F$ is naturally better-equipped to handle the compositionality of $\mathcal{F}$. In particular, we prove Lemma 2.1 by leveraging a uniform Lipschitz property of $m_F$. This uniform Lipschitz property does not hold for prior definitions of margin and reflects the key insight in our definition – it arises only because our margin depends on *simultaneous* perturbations to all layers.

**Claim 2.1.** *For any two compositions $F = f_k \circ \cdots \circ f_1$ and $\widehat{F} = \widehat{f}_k \circ \cdots \circ \widehat{f}_1$ and any $(x, y)$, we have $|m_F(x, y) - m_{\widehat{F}}(x, y)| \le \sqrt{\sum_{i=1}^k \|f_i - \widehat{f}_i\|_{\mathrm{op}}^2}$.*

*Proof sketch.* Let $\delta^\star$ be the optimal choice of $\delta$ in the definition of $m_F(x, y)$. We will construct $\widehat{\delta}$ such that $\|\widehat{\delta}\|_2 \le \|\delta^\star\|_2 + \sqrt{\sum_i \|f_i - \widehat{f}_i\|_{\mathrm{op}}^2}$ and $\gamma(\widehat{F}(x, \widehat{\delta}), y) = 0$ as follows: define $\widehat{\delta}_i \triangleq \delta_i^\star + \Delta_i$ for $\Delta_i \triangleq \frac{f_i(h_{i-1}(x, \delta^\star)) - \widehat{f}_i(h_{i-1}(x, \delta^\star))}{\|h_{i-1}(x, \delta^\star)\|_2}$, where $h$ is defined as in (2.1) with respect to the classifier $F$. Note that by our definition of $\|\cdot\|_{\mathrm{op}}$, we have $\|\Delta_i\|_2 \le \|f_i - \widehat{f}_i\|_{\mathrm{op}}$. Now it is possible to check inductively that $\widehat{F}(x, \widehat{\delta}) = F(x, \delta^\star)$. In particular, $\widehat{\delta}$ is satisfies the misclassification constraint in the all-layer margin objective for $\widehat{F}$. Thus, it follows that $m_{\widehat{F}}(x, y) \le \|\widehat{\delta}\|_2 \le \|\delta^\star\|_2 + \|\Delta\|_2 \le m_F(x, y) + \sqrt{\sum_i \|f_i - \widehat{f}_i\|_{\mathrm{op}}^2}$, where the last inequality followed from $\|\Delta_i\|_2 \le$

$\|f_i - \widehat{f_i}\|_{\text{op}}$. With the same reasoning, we obtain $m_F(x, y) \leq m_{\widehat{F}}(x, y) + \sqrt{\sum_i \|f_i - \widehat{f_i}\|^2_{\text{op}}}$, so $|m_F(x, y) - m_{\widehat{F}}(x, y)| \leq \sqrt{\sum_i \|f_i - \widehat{f_i}\|^2_{\text{op}}}$. $\qquad\square$

Given Claim 2.1, Lemma 2.1 follows simply by composing $\epsilon_i$-covers of $\mathcal{F}_i$. We prove a more general version in Section A (see Lemmas A.1 and A.3.)

The second part of the proof of Theorem 2.1 is to upper bound the 0-1 test error by the test error of some smooth surrogate loss $\ell \circ m_F$. A result by Srebro et al. (2010) shows that generic smooth losses $\ell$ enjoy faster $O(n^{-1})$ covergence rates if the empirical loss is low. We straightforwardly combine Lemma 2.1 with their results to obtain the following generalization bound for $\ell \circ m_F$:

**Lemma 2.2.** *Suppose that $\ell$ is a $\beta$-smooth loss function taking values in $[0, 1]$. Then in the setting of Theorem 2.1, we have with probability $1 - \delta$ for all $F \in \mathcal{F}$:*

$$\mathbb{E}_P[\ell(m_F(x, y))] \leq \frac{3}{2}\mathbb{E}_{P_n}[\ell(m_F(x, y))] + c\left(\frac{\beta(\sum_i \mathcal{C}_i)^2 \log^2 n}{n} + \frac{\log(1/\delta) + \log\log n}{n}\right) \quad (2.4)$$

*for some universal constant $c > 0$.*

To complete the proof of Theorem 2.1, we will choose $\ell \circ m_F$ which upper bounds the 0-1 loss such that the right hand side of (2.4) gives the desired bound. In Section A, we formalize the proof plan presented here and also define a slightly more general version of $m_F$ used to derive the bounds presented in the following Sections 3 and 4.

**Connection to (normalized) output margin** Finally, we check that when $F$ is a linear classifier, $m_F$ recovers the standard output margin. Thus, we can view the all-layer margin as an extension of the output margin to deeper classifiers.

**Example 2.1.** *In the binary classification setting with a linear classifier $F(x) = w^\top x$ where the data $x$ has norm 1, we have $m_F(x, y) = \max\{0, yw^\top x\} = \gamma(F(x), y)$.*

For deeper models, the all-layer margin can be roughly bounded by a quantity which normalizes the output margin by Jacobian and hidden layer norms. We formalize this in Lemma 3.1 and use this to prove our main generalization bound for neural nets, Theorem 3.1.

## 3 GENERALIZATION GUARANTEES FOR NEURAL NETWORKS

Although the all-layer margin is likely difficult to compute exactly, we can analytically lower bound it for neural nets with smooth activations. In this section, we obtain a generalization bound that depends on computable quantities by substituting this lower bound into Theorem 2.1. Our bound considers the Jacobian norms, hidden layer norms, and output margin on the training data, and avoids the exponential depth dependency when these quantities are well-behaved, as is the case in practice (Arora et al., 2018; Nagarajan and Kolter, 2019; Wei and Ma, 2019). Prior work (Nagarajan and Kolter, 2019; Wei and Ma, 2019) avoided the exponential depth dependency by considering these same quantities but required complicated proof frameworks. We obtain a simpler proof with tighter dependencies on these quantities by analyzing the all-layer margin.

The neural net classifier $F$ will be parameterized by $r$ weight matrices $\{W_{(i)}\}$ and compute $F(x) = W_{(r)}\phi(\cdots\phi(W_{(1)}x)\cdots)$ for smooth activation $\phi$. Let $d$ be the largest layer dimension. We model this neural net by a composition of $k = 2r - 1$ layers alternating between matrix multiplications and applications of $\phi$ and use the subscript in parenthesis $_{(i)}$ to emphasize the different indexing system between weight matrices and all the layers. We will let $s_{(i)}(x)$ denote the $\|\cdot\|_2$ norm of the layer preceding the $i$-th matrix multiplication evaluated on input $x$, and $\kappa_{j \leftarrow i}(x)$ will denote the $\|\cdot\|_{\text{op}}$ norm of the Jacobian of the $j$-th layer with respect to the $i - 1$-th layer evaluated on $x$. The following theorem bounds the generalization error of the network and is derived by lower bounding the all-layer margin in terms the quantities $s_{(i)}(x), \kappa_{j \leftarrow i}(x), \gamma(F(x), y)$.

**Theorem 3.1.** *Assume that the activation $\phi$ has a $\kappa'_\phi$-Lipschitz derivative. Fix reference matrices $\{A_{(i)}, B_{(i)}\}^k_{i=1}$ and any integer $q > 0$. With probability $1 - \delta$ over the draw of the training sample*

$P_n$, all neural nets $F$ which achieve training error 0 satisfy

$$\mathbb{E}_P[\ell_{0\text{-}1} \circ F] \leq O\left(\frac{\left(\sum_i \|\kappa_{(i)}^{\text{NN}}\|_{L_q(P_n)}^{2/3} a_{(i)}^{2/3}\right)^{3q/(q+2)} q \log^2 n}{n^{q/(q+2)}}\right) + \zeta \tag{3.1}$$

where $\kappa_{(i)}^{\text{NN}}$ captures a local Lipschitz constant of perturbations at layer $i$ and is defined by

$$\kappa_{(i)}^{\text{NN}}(x,y) \triangleq \frac{s_{(i-1)}(x)\kappa_{2r-1\leftarrow 2i}(x)}{\gamma(F(x),y)} + \psi_{(i)}(x,y) \tag{3.2}$$

for a secondary term $\psi_{(i)}(x,y)$ given by

$$\psi_{(i)}(x,y) \triangleq \sum_{j=i}^{r-1} \frac{s_{(i-1)}(x)\kappa_{2j\leftarrow 2i}(x)}{s_{(j)}(x)} + \sum_{1\leq j\leq 2i-1\leq j'\leq 2r-1} \frac{\kappa_{j'\leftarrow 2i}(x)\kappa_{2i-2\leftarrow j}(x)}{\kappa_{j'\leftarrow j}(x)}$$

$$+ \sum_{1\leq j\leq j'\leq 2r-1} \sum_{j''=\max\{2i,j\},j''\text{even}}^{j'} \frac{\kappa'_\phi \kappa_{j'\leftarrow j''+1}(x)\kappa_{j''-1\leftarrow 2i}(x)\kappa_{j''-1\leftarrow j}(x)s_{(i-1)}(x)}{\kappa_{j'\leftarrow j}(x)}$$

Here the second and third summations above are over the choices of $j, j'$ satisfying the constraints specified in the summation. We define $a_{(i)}$ by $a_{(i)} \triangleq \min\{\sqrt{d}\|W_{(i)} - A_{(i)}\|_{\text{fro}}, \|W_{(i)} - B_{(i)}\|_{1,1}\}\sqrt{\log d} + \text{poly}(n^{-1})$ and $\zeta \lesssim \frac{r \log n + \log(1/\delta) + \sum_i \log(a_{(i)}+1)}{n}$ is a low-order term.

For example, when $q = 2$, from (3.1) we obtain the following bound which depends on the second moment of $\kappa_{(i)}^{\text{NN}}$ and features the familiar $1/\sqrt{n}$ convergence rate in the training set size.

$$\mathbb{E}_P[\ell_{0\text{-}1} \circ F] \lesssim \frac{\left(\sum_i \mathbb{E}_{P_n}\left[(\kappa_{(i)}^{\text{NN}})^2\right]^{1/3}(a_{(i)})^{2/3}\right)^{3/2} \log^2 n}{\sqrt{n}} + \xi$$

For larger $q$, we obtain a faster convergence rate in $n$, but the dependency on $\kappa_{(i)}^{\text{NN}}$ gets larger. We will outline a proof sketch which obtains a variant of Theorem 3.1 with a slightly worse polynomial dependency on $\kappa_{(i)}^{\text{NN}}$ and $a_{(i)}$. For simplicity we defer the proof of the full Theorem 3.1 to Sections B and C. First, we need to slightly redefine $m_F$ so that perturbations are only applied at linear layers (formally, fix $\delta_{2i} = 0$ for the even-indexed activation layers, and let $\delta_{(i)} \triangleq \delta_{2i-1}$ index perturbations to the $i$-th linear layer). It is possible to check that Lemma 2.1 still holds since activation layers correspond to a singleton function class $\{\phi\}$ with log covering number 0. Thus, the conclusion of Theorem 2.1 also applies for this definition of $m_F$. Now the following lemma relates this all-layer margin to the output margin and Jacobian and hidden layer norms, showing that $m_F(x,y)$ can be lower bounded in terms of $\{\kappa_{(i)}^{\text{NN}}(x,y)\}$.

**Lemma 3.1.** *In the setting above, we have the lower bound* $m_F(x,y) \geq \frac{1}{\|\{\kappa_{(i)}^{\text{NN}}(x,y)\}_{i=1}^r\|_2}$.

Directly plugging the above lower bound into Theorem 2.1 and choosing $\mathcal{C}_{2i} = 0$, $\mathcal{C}_{2i-1} = a_{(i)}$ would give a variant of Theorem 3.1 that obtains a different polynomial in $\kappa_{(i)}^{\text{NN}}$, $a_{(i)}$.

**Heuristic derivation of Lemma 3.1** We compute the derivative of $F(x,\delta)$ with respect to $\delta_{(i)}$: $D_{\delta_{(i)}}F(x,\delta) = D_{h_{2i-1}(x,\delta)}F(x,\delta)\|h_{2i-2}(x,\delta)\|_2$. We abused notation to let $D_{h_{2i-1}(x,\delta)}F(x,\delta)$ denote the derivative of $F$ with respect to the $2i - 1$-th perturbed layer evaluated on input $(x,\delta)$. By definitions of $\kappa_{j\leftarrow i}$, $s_{(i)}$ and the fact that the output margin is 1-Lipschitz, we obtain $\|D_{\delta_{(i)}}\gamma(F(x,\delta),y)|_{\delta=0}\|_2 \leq \|D_{h_{2i-1}(x,0)}F(x,0)\|_{\text{op}}\|h_{2i-2}(x,0)\|_2 = \kappa_{2r-1\leftarrow 2i}(x)s_{(i-1)}(x)$. With the first order approximation $\gamma(F(x,\delta),y) \approx \gamma(F(x),y) + \sum_i D_{\delta_{(i)}}\gamma(F(x,\delta),y)|_{\delta=0}\delta_{(i)}$ around $\delta = 0$, we obtain

$$\gamma(F(x,\delta),y) \geq \gamma(F(x),y) - \sum_i \kappa_{2r-1\leftarrow 2i}(x)s_{(i-1)}(x)\|\delta_{(i)}\|_2$$

The right hand side is nonnegative whenever $\|\delta\|_2 \leq \frac{\gamma(F(x),y)}{\|\{\kappa_{2r-1\leftarrow 2i}(x)s_{(i-1)}(x)\}_{i=1}^r\|_2}$, which would imply that $m_F(x,y) \geq \frac{\gamma(F(x),y)}{\|\{\kappa_{2r-1\leftarrow 2i}(x)s_{(i-1)}(x)\}_{i=1}^r\|_2}$. However, this conclusion is imprecise and

non-rigorous because of the first order approximation – to make the argument rigorous, we also control the smoothness of the network around $x$ in terms of the interlayer Jacobians, ultimately resulting in the bound of Lemma 3.1. We remark that the quantities $\kappa_{(i)}^{\mathrm{NN}}$ are not the only expressions with which we could lower bound $m_F(x, y)$. Rather, the role of $\kappa_{(i)}^{\mathrm{NN}}$ is to emphasize the key term $\frac{s_{(i-1)}(x)\kappa_{2r-1\leftarrow 2i}(x)}{\gamma(F(x),y)}$, which measures the first order stability of the network to perturbation $\delta_{(i)}$ and relates the all-layer margin to the output margin. As highlighted above, if this term is small, $m_F$ will be large so long as the network is sufficiently smooth around $(x, y)$, as captured by $\psi_{(i)}(x, y)$.

**Comparison to existing bounds**  We can informally compare Theorem 3.1 to the existing bounds of (Nagarajan and Kolter, 2019; Wei and Ma, 2019) as follows. First, the leading term $\frac{s_{(i-1)}(x)\kappa_{2r-1\leftarrow 2i}(x)}{\gamma(F(x),y)}$ of $\kappa_{(i)}^{\mathrm{NN}}$ depends on three quantities all evaluated *on the same training example*, whereas the analogous quantity in the bounds of (Nagarajan and Kolter, 2019; Wei and Ma, 2019) appears as $\max_{P_n} \frac{1}{\gamma(F(x),y)} \cdot \max_{P_n} s_{(i-1)}(x) \cdot \max_{P_n} \kappa_{2r-1\leftarrow 2i}(x)$, where each maximum is taken over the entire training set. As we have $\|\kappa_{(i)}^{\mathrm{NN}}\|_{L_q(P_n)} \leq \max_{P_n} \kappa_{(i)}^{\mathrm{NN}}(x, y)$, the term $\|\kappa_{(i)}^{\mathrm{NN}}\|_{L_q(P_n)}$ in our bound can be much smaller than its counterpart in the bounds of (Nagarajan and Kolter, 2019; Wei and Ma, 2019). An interpretation of the parameter $q$ is that we obtain fast (close to $n^{-1}$) convergence rates if the model fits every training example perfectly with large all-layer margin, or we could have slower convergence rates with better dependence on the all-layer margin distribution. It is unclear whether the techniques in other papers can achieve convergence rates faster than $O(1/\sqrt{n})$ because their proofs require the simultaneous convergence of multiple data-dependent quantities, whereas we bound everything using the single quantity $m_F$.

Additionally, we compare the dependence on the weight matrix norms relative to $n$ (as the degree of $n$ in our bound can vary). For simplicity, assume that the reference matrices $A_{(i)}$ are set to $0$. Our dependence on the weight matrix norms relative to the training set size is, up to logarithmic factors, $\left( \frac{\min\{\sqrt{d}\|W_{(i)}\|_{\mathrm{fro}}, \|W_{(i)}\|_{1,1}\}}{\sqrt{n}} \right)^{2q/(q+2)}$, which always matches or improves on the dependency obtained by PAC-Bayes methods such as (Neyshabur et al., 2017b; Nagarajan and Kolter, 2019). Wei and Ma (2019) obtain the dependency $\frac{\|W_{(i)}^{\top}\|_{2,1}}{\sqrt{n}}$, where $\|W_{(i)}^{\top}\|_{2,1}$ is the sum of the $\|\cdot\|_2$ norms of the rows of $W_{(i)}$. This dependency on $W_{(i)}$ is always smaller than ours. Finally, we note that Theorem 2.1 already gives tighter (but harder to compute) generalization guarantees for relu networks directly in terms of $m_F$. Existing work contains a term which depends on inverse pre-activations shown to be large in practice (Nagarajan and Kolter, 2019), whereas $m_F$ avoids this dependency and is potentially much smaller. We explicitly state the bound in Section B.1.

## 4  GENERALIZATION GUARANTEES FOR ROBUST CLASSIFICATION

In this section, we apply our tools to obtain generalization bounds for adversarially robust classification. Prior works rely on relaxations of the adversarial loss to bound adversarially robust generalization for neural nets (Khim and Loh, 2018; Yin et al., 2018). These relaxations are not tight and in the case of (Yin et al., 2018), only hold for neural nets with one hidden layer. To the best of our knowledge, our work is the first to *directly* bound generalization of the robust classification error for any network. Our bounds are formulated in terms of data-dependent properties in the adversarial neighborhood of the training data and avoid explicit exponential dependencies in depth. Let $\mathcal{B}^{\mathrm{adv}}(x)$ denote the set of possible perturbations to the point $x$ (typically some norm ball around $x$). We would like to bound generalization of the adversarial classification loss $\ell_{0\text{-}1}^{\mathrm{adv}}$ defined by $\ell_{0\text{-}1}^{\mathrm{adv}}(F(x), y) \triangleq \max_{x' \in \mathcal{B}^{\mathrm{adv}}(x)} \ell_{0\text{-}1}(F(x'), y)$. We prove the following bound which essentially replaces all data-dependent quantities in Theorem 3.1 with their adversarial counterparts.

**Theorem 4.1.** *Assume that the activation $\phi$ has a $\kappa_{\phi}'$-Lipschitz derivative. Fix reference matrices $\{A_{(i)}, B_{(i)}\}_{i=1}^{k}$ and any integer $q > 0$. With probability $1 - \delta$ over the draw of the training sample $P_n$, all neural nets $F$ which achieve robust training error 0 satisfy*

$$\mathbb{E}_P[\ell_{0\text{-}1}^{\mathrm{adv}} \circ F] \leq O\left( \frac{q \log^2 n \left( \sum_i \|\kappa_{(i)}^{\mathrm{adv}}\|_{L_q(P_n)}^{2/3} a_{(i)}^{2/3} \right)^{3q/(q+2)}}{n^{q/(q+2)}} \right) + \zeta$$

where $\kappa_{(i)}^{\mathrm{adv}}$ is defined by $\kappa_{(i)}^{\mathrm{adv}}(x, y) \triangleq \max_{x' \in \mathcal{B}^{\mathrm{adv}}(x)} \kappa_{(i)}^{\mathrm{NN}}(x', y)$ for $\kappa_{(i)}^{\mathrm{NN}}$ in (3.2), and $a_{(i)}, \zeta$ are defined the same as in Theorem 3.1.

Designing regularizers for robust classification based on the bound in Theorem 4.1 is a promising direction for future work. To prove Theorem 4.1, we simply define a natural extension to our all-layer margin, and the remaining steps follow in direct analogy to the clean classification setting. We define the adversarial all-layer margin as the smallest all-layer margin on the perturbed inputs: $m_F^{\mathrm{adv}}(x, y) \triangleq \min_{x' \in \mathcal{B}^{\mathrm{adv}}(x)} m_F(x, y)$. We note that $m_F^{\mathrm{adv}}(x, y)$ is nonzero if and only if $F$ correctly classifies all adversarial perturbations of $x$. Furthermore, the adversarial all-layer margin also satisfies the uniform Lipschitz property in Claim 2.1. Thus, the remainder of the proof of Theorem 4.1 follows the same steps laid out in Section 2. As before, we note that Theorem 4.1 requires $m_F$ to be the more general all-layer margin defined in Section A. We provide the full proofs in Section E.

## 5 EMPIRICAL APPLICATION OF THE ALL-LAYER MARGIN

Inspired by the good statistical properties of the all-layer margin, we design an algorithm which encourages a larger all-layer margin during training. Letting $\ell$ denote the standard cross entropy loss used in training and $\Theta$ the parameters of the network, consider the following objective:

$$G(\delta, \Theta; x, y) \triangleq \ell(F_\Theta(x, \delta), y) - \lambda \|\delta\|_2^2 \tag{5.1}$$

This objective can be interpreted as applying the Lagrange multiplier method to a softmax relaxation of the constraint $\max_{y'} F(x, \delta_1, \ldots, \delta_k)_{y'} \neq y$ in the objective for all-layer margin. If $G(\delta, \Theta; x, y)$ is large, this signifies the existence of some $\delta$ with small norm for which $F_\Theta(x, \delta)$ suffers large loss, indicating that $m_{F_\Theta}$ is likely small. This motivates the following training objective over $\Theta$: $L(\Theta) \triangleq \mathbb{E}_{P_n}[\max_\delta G(\delta, \Theta; x, y)]$. Define $\delta_{\Theta, x, y}^\star \in \arg\max_\delta G(\delta, \Theta; x, y)$. From Danskin's Theorem, if $G(\delta, \Theta; x, y)$ is continuously differentiable[2], then we have that the quantity $-\mathbb{E}_{P_n}[\nabla_\Theta G(\delta_{\Theta, x, y}^\star, \Theta; x, y)]$ will be a descent direction in $\Theta$ for the objective $L(\Theta)$ (see Corollary A.2 of (Madry et al., 2017) for the derivation of a similar statement). Although the exact value $\delta_{\Theta, x, y}^\star$ is hard to obtain, we can use a substitute $\tilde{\delta}_{\Theta, x, y}$ found via several gradient ascent steps in $\delta$. This inspires the following all-layer margin optimization (AMO) algorithm: we find perturbations $\tilde{\delta}$ for each example in the batch via gradient ascent steps on $G(\delta, \Theta; x, y)$. For each example in the batch, we then compute the perturbed loss $\ell(F_\Theta(x, \tilde{\delta}_{\Theta, x, y}))$ and update $\Theta$ with its negative gradient with respect to these perturbed losses. This method is formally outlined in the PERTURBEDUPDATE procedure of Algorithm 1.[3]

We use Algorithm 1 to train a WideResNet architecture (Zagoruyko and Komodakis, 2016) on CIFAR10 and CIFAR100 in a variety of settings. For all of our experiments we use $t = 1$, $\eta_{\mathrm{perturb}} = 0.01$, and we apply perturbations following conv layers in the WideResNet basic blocks. In Table 1 we report the best validation error achieved during a single run of training, demonstrating that our algorithm indeed leads to improved generalization over the strong WideResNet baseline for a variety of settings. Additional parameter settings and experiments for the feedforward VGG (Simonyan and Zisserman, 2014) architecture are in Section F.1. In addition, in Section F.1, we show that dropout, another regularization method which perturbs each hidden layer, does not offer the same improvement as AMO.

Inspired by our robust generalization bound, we also apply AMO to robust classification by extending the robust training algorithm of (Madry et al., 2017). Madry et al. (2017) adversarially perturb the training input via several steps of projected gradient descent (PGD) and train using the loss computed on this perturbed input. At each update, our robust AMO algorithm initializes perturbations $\delta$ to 0 for every training example in the batch. The updates to these $\delta$ are performed simultaneously with PGD updates for the adversarial perturbations with the same update rule as Algorithm 1.

---

[2] If we use a relu network, $G(\delta, \Theta; x, y)$ is technically not continuously differentiable, but the algorithm that we derive still works empirically for relu nets.

[3] We note a slight difference with our theory: in the FORWARDPERTURB function, we perform the update $f_{j \leftarrow 1}(x, \delta) = f_j(f_{j-1 \leftarrow 1}(x, \delta)) + \|f_j(f_{j-1 \leftarrow 1}(x, \delta))\| \delta_j$, rather than scaling $\delta$ by the previous layer norm – this allows the perturbation to also account for the scaling of layer $j$.

---

**Algorithm 1** All-layer Margin Optimization (AMO)

---

**procedure** PERTURBEDUPDATE(minibatch $B = \{(x_i, y_i)\}_{i=1}^b$, current parameters $\Theta$)

    Initialize $\delta_i = 0$ for $i = 1, \ldots, b$.

    **for** $s = 1, \ldots, t$ **do**

        **for all** $(x_i, y_i) \in B$: **do**

            Update $\delta_i \leftarrow (1 - \eta_{\text{perturb}}\lambda)\delta_i + \eta_{\text{perturb}}\nabla_\delta \ell(\text{FORWARDPERTURB}(x_i, \delta_i, \Theta), y_i)$

    Set update $g = \nabla_\Theta \left[\frac{1}{b}\sum_i \ell(\text{ FORWARDPERTURB}(x_i, \delta_i, \Theta), y_i)\right]$.

    Update $\Theta \leftarrow \Theta - \eta(g + \nabla_\Theta R(\Theta))$.           $\triangleright$ $R$ is a regularizer, i.e. weight decay.

**function** FORWARDPERTURB($x, \delta, \Theta$)         $\triangleright$ The net has layers $f_1(\cdot; \Theta), \ldots, f_r(\cdot; \Theta)$,

                                      with intended perturbations $\delta^{(1)}, \ldots, \delta^{(r)}$.

    Initialize $h \leftarrow x$.

    **for** $j = 1, \ldots, r$ **do**

        Update $h \leftarrow f_j(h; \Theta)$.

        Update $h \leftarrow h + \|h\|\delta^{(j)}$.

    **return** $h$

---

Table 1: Validation error on CIFAR for standard training vs. AMO (Algorithm 1).

| Dataset | Arch. | Setting | Standard SGD | AMO |
|---------|-------|---------|--------------|-----|
| CIFAR-10 | WRN16-10 | Baseline | 4.15% | **3.42%** |
| | | No data augmentation | 9.59% | **6.74%** |
| | | 20% random labels | 9.43% | **6.72%** |
| | WRN28-10 | Baseline | 3.82% | **3.00%** |
| | | No data augmentation | 8.28% | **6.47%** |
| | | 20% random labels | 8.17% | **6.01%** |
| CIFAR-100 | WRN16-10 | Baseline | 20.12% | **19.14%** |
| | | No data augmentation | 31.94% | **26.09%** |
| | WRN28-10 | Baseline | 18.85% | **17.78%** |
| | | No data augmentation | 30.04% | **24.67%** |

In Table 2, we report best validation performance for robust AMO and the baseline method of (Madry et al., 2017) on CIFAR10. Our results demonstrate that our robust AMO algorithm can also offer improvements in the robust accuracy. We provide parameter settings in Section F.2.

## 6 CONCLUSION

Many popular objectives in deep learning are based on maximizing a notion of output margin, but unfortunately it is difficult to obtain good statistical guarantees by analyzing this output margin. In this paper, we design a new all-layer margin which attains strong statistical guarantees for deep models. Our proofs for these guarantees follow very naturally from our definition of the margin. We apply the all-layer margin in several ways: 1) we obtain tighter data-dependent generalization bounds for neural nets 2) for adversarially robust classification, we directly bound the robust generalization error in terms of local Lipschitzness around the perturbed training examples, and 3) we design a new algorithm to encourage larger all-layer margins and demonstrate improved performance on real data in both clean and adversarially robust classification settings. We hope that our results prompt further study on maximizing all-layer margin as a new objective for deep learning.

Table 2: Robust validation error on CIFAR-10 for standard robust training (Madry et al., 2017) vs. robust AMO. The attack model is 50 steps of PGD with 10 random restarts using $\ell_\infty$ perturbations with radius $\epsilon = 8$.

| Arch. | (Madry et al., 2017) | Robust AMO |
|-------|----------------------|------------|
| WideResNet16-10 | 48.75% | **45.94%** |
| WideResNet28-10 | 45.47% | **42.38%** |

## 7 ACKNOWLEDGEMENTS

CW acknowledges support from an NSF Graduate Research Fellowship. The work is also partially supported by SDSI and SAIL.

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

## A    GENERALIZED ALL-LAYER MARGIN AND MISSING PROOFS FOR SECTION 2

In this section, we provide proofs for Section 2 in a more general and rigorous setting. We first formally introduce the setting, which considers functions composed of layers which map between arbitrary normed spaces.

Recall that $F$ denotes our classifier from Section 2 computed via the composition $f_k \circ \cdots \circ f_1$, For convenience, we overload notation and also let it refer to the sequence of functions $\{f_1, \ldots, f_k\}$. Recall that $\mathcal{F}$ denotes the class of all compositions of layers $\mathcal{F}_k, \ldots, \mathcal{F}_1$, where we let functions in $\mathcal{F}_i$ map domains $\mathcal{D}_{i-1}$ to $\mathcal{D}_i$. We will fix $\mathcal{D}_k \triangleq \mathbb{R}^l$, the space of predictions for $l$ classes. Each space is equipped with norm $\|\cdot\|$ (our theory allows the norm to be different for every $i$, but for simplicity we use the same symbol $\|\cdot\|$ for all layers).

As in Section 2, we will use $F(x, \delta)$ to denote the classifier output perturbed by $\delta$. It will be useful to define additional notation for the perturbed function between layers $i$ and $j$, denoted by $f_{j \leftarrow i}(h, \delta)$, recursively as follows:

$$f_{i \leftarrow i}(h, \delta) \triangleq f_i(h) + \delta_i \|h\|, \text{ and } f_{j \leftarrow i}(h, \delta) \triangleq f_{j \leftarrow j}(f_{j-1 \leftarrow i}(h, \delta)) + \delta_j \|f_{j-1 \leftarrow i}(h, \delta)\|$$

where we choose $f_{i-1 \leftarrow i}(h, \delta) \triangleq h$. Note that $F(x, \delta) \triangleq f_{k \leftarrow 1}(x, \delta)$, and the notation $h_i(x, \delta)$ from Section 2 is equivalent to $f_{i \leftarrow 1}(x, \delta)$. We will use the simplified notation $f_{j \leftarrow i}(x) \triangleq f_{j \leftarrow i}(x, 0)$ when the perturbation $\delta$ is 0 at all layers.

For a given $F$, we now define the general all-layer margin $m_F : \mathcal{D}_0 \times [l] \to \mathbb{R}$ as follows:

$$m_F(x, y) \triangleq \begin{matrix} \min \||\delta\|| \\ \text{subject to } \gamma(F(x, \delta), y) \le 0 \end{matrix} \tag{A.1}$$

The norm $\||\cdot\||$ will have the following form:

$$\||\delta\|| = \|(\alpha_1 \|\delta_1\|, \ldots, \alpha_k \|\delta_k\|)\|_p$$

where $\alpha_i \ge 0$ will be parameters chosen later to optimize the resulting bound, and $\|\cdot\|_p$ denotes the standard $\ell_p$-norm. For $F = \{f_1, \ldots, f_k\}$, we overload notation and write $\||F\|| = \|(\alpha_1 \|f_1\|_{\text{op}}, \ldots, \alpha_k \|f_k\|_{\text{op}})\|_p$.

This more general definition of $m_F$ will be useful for obtaining Theorems 3.1 and 4.1. Note that by setting $\alpha_i = 1$ for all $i$ and $p = 2$, we recover the simpler $m_F$ defined in Section 2.

As before, it will be convenient for the analysis to assume that the $\epsilon$-covering number of $\mathcal{F}_i$ in operator norm scales with $\epsilon^{-2}$. We formally state this condition for general function classes and norms below:

**Condition A.1** ($\epsilon^{-2}$ covering condition). *We say that a function class $\mathcal{G}$ satisfies the $\epsilon^{-2}$ covering condition with respect to norm $\|\cdot\|$ with complexity $\mathcal{C}_{\|\cdot\|}(\mathcal{G})$ if for all $\epsilon > 0$,*

$$\log \mathcal{N}_{\|\cdot\|}(\epsilon, \mathcal{G}) \le \left\lfloor \frac{\mathcal{C}_{\|\cdot\|}^2(\mathcal{G})}{\epsilon^2} \right\rfloor$$

Now we provide the analogue of Theorem 2.1 for the generalized all-layer margin:

**Theorem A.1.** *Fix any integer $q > 0$. Suppose that all layer functions $\mathcal{F}_i$ satisfy Condition A.1 with operator norm $\|\cdot\|_{\text{op}}$ and complexity function $\mathcal{C}_{\|\cdot\|_{\text{op}}}(\mathcal{F}_i)$. Let the all layer margin $m_F$ be defined as in (A.1). Then with probability $1 - \delta$ over the draw of the training data, all classifiers $F \in \mathcal{F}$ which achieve training error 0 satisfy*

$$\mathbb{E}_P[\ell_{\text{0-1}}(F(x), y)] \lesssim \left( \frac{\left\| \frac{1}{m_F} \right\|_{L_q(P_n)} \mathcal{C}_{\|\cdot\||}(\mathcal{F})}{\sqrt{n}} \right)^{2q/(q+2)} q \log^2 n + \zeta$$

*where $\mathcal{C}_{\|\cdot\||}(\mathcal{F}) \triangleq \left( \sum_i \alpha_i^{2p/(p+2)} \mathcal{C}_{\|\cdot\|_{\text{op}}}(\mathcal{F}_i)^{2p/(p+2)} \right)^{(p+2)/2p}$ is a complexity (in the sense of Condition A.1) for covering $\mathcal{F}$ in $\|\cdot\||$ and $\zeta \triangleq O\left( \frac{\log(1/\delta) + \log n}{n} \right)$ is a low-order term.*

Note that this recovers Theorem 2.1 when $\alpha_i = 1$ for all $i$ and $p = 2$. The proof of Theorem A.1 mirrors the plan laid out in Section 2. As before, the first step of the proof is showing that $m_F$ has low complexity as measured by covering numbers.

**Lemma A.1.** *Define* $m \circ \mathcal{F} \triangleq \{(x, y) \mapsto m_F(x, y) : F \in \mathcal{F}\}$. *Then*

$$\mathcal{N}_\infty(\epsilon, m \circ \mathcal{F}) \leq \mathcal{N}_{\|\cdot\|}(\epsilon, \mathcal{F})$$

As in Section 2, we prove Lemma A.1 by bounding the error between $m_F$ and $m_{\widehat{F}}$ in terms of the $\|\cdot\|$-norm of the difference between $F$ and $\widehat{F}$.

**Claim A.1.** *For any* $x, y \in \mathcal{D}_0 \times [l]$, *and function sequences* $F = \{f_i\}_{i=1}^k$, $\widehat{F} = \{\widehat{f}_i\}_{i=1}^k$, *we have* $|m_F(x, y) - m_{\widehat{F}}(x, y)| \leq \|F - \widehat{F}\|$.

*Proof.* Suppose that $\delta^\star$ optimizes equation (A.1) used to define $m_F(x, y)$. Now we use the notation $h_i^\star \triangleq f_{i \leftarrow 1}(x, \delta^\star)$. Define $\widehat{\delta}$ as follows:

$$\widehat{\delta}_i \triangleq \delta_i^\star + \frac{f_i(h_{i-1}^\star) - \widehat{f}_i(h_{i-1}^\star)}{\|h_{i-1}^\star\|}$$

We first argue via induction that $\widehat{f}_{i \leftarrow 1}(x, \widehat{\delta}) = h_i^\star$. As the base case, we trivially have $\widehat{f}_{0 \leftarrow 1}(x, \widehat{\delta}) = x = h_0^\star$.

$$
\begin{aligned}
\widehat{f}_{i \leftarrow 1}(x, \widehat{\delta}) &= \widehat{f}_i(\widehat{f}_{i-1 \leftarrow 1}(x, \widehat{\delta})) + \widehat{\delta}_i \|\widehat{f}_{i-1 \leftarrow 1}(x, \widehat{\delta})\| \\
&= \widehat{f}_i(h_{i-1}^\star) + \widehat{\delta}_i \|h_{i-1}^\star\| && \text{(by the inductive hypothesis)} \\
&= \widehat{f}_i(h_{i-1}^\star) + \left(\delta_i^\star + \frac{f_i(h_{i-1}^\star) - \widehat{f}_i(h_{i-1}^\star)}{\|h_{i-1}^\star\|}\right) \|h_{i-1}^\star\| && \text{(definition of } \widehat{\delta}) \\
&= f_i(h_{i-1}^\star) + \delta_i^\star \|h_{i-1}^\star\| \\
&= h_i^\star && \text{(definition of } g_i^\star)
\end{aligned}
$$

Thus, we must have $\widehat{F}(x, \widehat{\delta}) = F(x, \delta^\star)$, so it follows that $\gamma(\widehat{F}(x, \widehat{\delta}), y) \leq 0$ as well. Furthermore, by triangle inequality

$$\|\widehat{\delta}\| \leq \|\delta^\star\| + \|\widehat{\delta} - \delta^\star\| \tag{A.2}$$

Now we note that as $\frac{\|f_i(h_{i-1}^\star) - \widehat{f}_i(h_{i-1}^\star)\|}{\|h_{i-1}^\star\|} \leq \|f_i - \widehat{f}_i\|_{\mathrm{op}}$, it follows that

$$\|\widehat{\delta} - \delta^\star\| \leq \|(\alpha_1 \|f_1 - \widehat{f}_1\|_{\mathrm{op}}, \ldots, \alpha_k \|f_k - \widehat{f}_k\|_{\mathrm{op}})\|_p = \|F - \widehat{F}\|$$

Thus, using (A.2) and the definition of $m_{\widehat{F}}$, we have

$$m_{\widehat{F}}(x, y) \leq \|\widehat{\delta}\| \leq m_F(x, y) + \|F - \widehat{F}\|$$

where we relied on the fact that $\|\delta^\star\| = m_F(x, y)$. Using the same reasoning, we also obtain the inequality $m_F(x, y) \leq m_{\widehat{F}}(x, y) + \|F - \widehat{F}\|$. Combining gives $|m_F(x, y) - m_{\widehat{F}}(x, y)| \leq \|F - \widehat{F}\|$. $\qquad\square$

Lemma A.1 now directly follows.

*Proof of Lemma A.1.* As Claim A.1 holds for any choice of $(x, y) \in \mathcal{D}_0 \times [l]$, it follows that if $\widehat{\mathcal{F}}$ covers $\mathcal{F}$ in norm $\|\cdot\|$, then $m \circ \widehat{\mathcal{F}}$ will be a cover for $m \circ \mathcal{F}$ in the functional $\infty$ norm. $\qquad\square$

We now state the generalized version of Lemma 2.2. The statement below is a straightforward application of our covering number bound in Lemma A.1 with theory in (Srebro et al., 2010); for minor technical reasons we translate their result to covering numbers and reprove it in Section A.1.

**Lemma A.2** (Straightforward adaptation from (Srebro et al., 2010))**.** *Suppose that $\ell$ is a $\beta$-smooth loss function taking values in $[0, 1]$. Furthermore suppose that $\mathcal{F}$ satisfies Condition A.1 with respect to norm $\|\cdot\|$ and complexity $\mathcal{C}_{\|\cdot\|}(\mathcal{F})$. Then with probability $1 - \delta$, for all $F \in \mathcal{F}$,*

$$\mathbb{E}_P[\ell(m_F(x, y))] \le \frac{3}{2}\mathbb{E}_{P_n}[\ell(m_F(x, y))] + c\left(\frac{\beta\mathcal{C}_{\|\cdot\|}^2(\mathcal{F})\log^2 n}{n} + \frac{\log(1/\delta) + \log\log n}{n}\right)$$

*for some universal constant $c > 0$.*

The final ingredient is showing that when each individual layer $\mathcal{F}_i$ satisfies Condition A.1 in operator norm, the class of compositions $\mathcal{F}$ satisfies Condition A.1 with respect to norm $\|\cdot\|$.

**Lemma A.3.** *Suppose that each $\mathcal{F}_i$ satisfies Condition A.1 with norm $\|\cdot\|_{\mathrm{op}}$ and complexity $\mathcal{C}_{\|\cdot\|_{\mathrm{op}}}(\mathcal{F}_i)$. Define the complexity measure $\mathcal{C}_{\|\cdot\|}(\mathcal{F})$ by*

$$\mathcal{C}_{\|\cdot\|}(\mathcal{F}) \triangleq \left(\sum_i \alpha_i^{2p/(p+2)}\mathcal{C}_{\|\cdot\|_{\mathrm{op}}}(\mathcal{F}_i)^{2p/(p+2)}\right)^{(p+2)/2p} \tag{A.3}$$

*Then we have*

$$\log\mathcal{N}_{\|\cdot\|}(\epsilon, \mathcal{F}) \le \left\lfloor \frac{\mathcal{C}_{\|\cdot\|}^2(\mathcal{F})}{\epsilon^2} \right\rfloor$$

*which by definition implies that $\mathcal{F}$ satisfies Condition A.1 with norm $\|\cdot\|$ and complexity $\mathcal{C}_{\|\cdot\|}(\mathcal{F})$.*

*Proof.* Let $\widehat{\mathcal{F}}_i$ be an $\epsilon_i$-cover of $\mathcal{F}_i$ in the operator norm $\|\cdot\|_{\mathrm{op}}$. We will first show that $\widehat{\mathcal{F}} \triangleq \{\widehat{F}_k \circ \cdots \circ \widehat{F}_1 : \widehat{F}_i \in \widehat{\mathcal{F}}_i\}$ is a $\|\{\alpha_i \epsilon_i\}_{i=1}^k\|_p$-cover of $\mathcal{F}$ in $\|\cdot\|$. To see this, for any $F = (f_k, \ldots, f_1) \in \mathcal{F}$, let $\widehat{f}_i \in \widehat{\mathcal{F}}_i$ be the cover element for $f_i$, and define $\widehat{F} \triangleq (\widehat{f}_k, \ldots, \widehat{f}_1)$. Then we have

$$\|\widehat{F} - F\| = \|\{\alpha_i\|\widehat{f}_i - f_i\|_{\mathrm{op}}\}_{i=1}^k\|_p$$
$$\le \|\{\alpha_i\epsilon_i\}_{i=1}^k\|_p$$

as desired. Furthermore, we note that $\log|\widehat{\mathcal{F}}| \le \sum_i \left\lfloor \frac{\mathcal{C}_{\|\cdot\|_{\mathrm{op}}}^2(\mathcal{F}_i)}{\epsilon_i^2} \right\rfloor$ by Condition A.1. Now we will choose

$$\epsilon_i = \epsilon\mathcal{C}_{\|\cdot\|}(\mathcal{F})^{-2/(p+2)}\mathcal{C}_{\|\cdot\|_{\mathrm{op}}}(\mathcal{F}_i)^{2/(p+2)}\alpha_i^{-p/(p+2)}$$

We first verify that this gives an $\epsilon$-cover of $\mathcal{F}$ in $\|\cdot\|$:

$$\|\{\alpha_i\epsilon_i\}_{i=1}^k\|_p = \epsilon\left(\mathcal{C}_{\|\cdot\|}(\mathcal{F})^{-2p/(p+2)}\sum_i \mathcal{C}_{\|\cdot\|_{\mathrm{op}}}(\mathcal{F}_i)^{2p/(p+2)}\alpha_i^{p-p^2/(p+2)}\right)^{1/p}$$

$$= \epsilon\mathcal{C}_{\|\cdot\|}(\mathcal{F})^{-2/(p+2)}\left(\sum_i \mathcal{C}_{\|\cdot\|_{\mathrm{op}}}(\mathcal{F}_i)^{2p/(p+2)}\alpha_i^{2p/(p+2)}\right)^{1/p}$$

$$= \epsilon\mathcal{C}_{\|\cdot\|}(\mathcal{F})^{-2/(p+2)}\mathcal{C}_{\|\cdot\|}(\mathcal{F})^{2/(p+2)} = \epsilon$$

Next, we check that the covering number is bounded by $\left\lfloor \frac{\mathcal{C}_{\|\cdot\|}^2(\mathcal{F})}{\epsilon^2} \right\rfloor$:

$$\sum_i \left\lfloor \frac{\mathcal{C}_{\|\cdot\|_{\mathrm{op}}}^2(\mathcal{F}_i)}{\epsilon_i^2} \right\rfloor \le \left\lfloor \frac{\sum_i \mathcal{C}_{\|\cdot\|_{\mathrm{op}}}^2(\mathcal{F}_i)^2\mathcal{C}_{\|\cdot\|}(\mathcal{F})^{4/(p+2)}\mathcal{C}_{\|\cdot\|_{\mathrm{op}}}(\mathcal{F}_i)^{-4/(p+2)}\alpha_i^{2p/(p+2)}}{\epsilon^2} \right\rfloor$$

$$= \left\lfloor \frac{\sum_i \mathcal{C}_{\|\cdot\|}(\mathcal{F})^{4/(p+2)}\mathcal{C}_{\|\cdot\|_{\mathrm{op}}}(\mathcal{F}_i)^{2p/(p+2)}\alpha_i^{2p/(p+2)}}{\epsilon^2} \right\rfloor$$

$$= \left\lfloor \frac{\mathcal{C}_{\|\cdot\|}(\mathcal{F})^{4/(p+2)}\mathcal{C}_{\|\cdot\|}(\mathcal{F})^{2p/(p+2)}}{\epsilon^2} \right\rfloor = \left\lfloor \frac{\mathcal{C}_{\|\cdot\|}(\mathcal{F})^2}{\epsilon^2} \right\rfloor$$

$\square$

Finally, we prove Theorem A.1 (and as a result, Theorem 2.1). This will hinge on applying Lemma A.2 with the correct choice of smooth loss.

*Proof of Theorems A.1 and 2.1.* Define $\ell_\beta(m) \triangleq 1 + 2\min\{0, \Pr_{Z \sim \mathcal{N}(0,1)}(Z/\sqrt{\beta} \geq m) - 0.5\}$. By Claim A.2, this loss is $c_1\beta$ smooth and for $m_F(x, y) > 0$ satisfies

$$\ell_\beta(m_F(x, y)) \leq \left(\frac{c_2\sqrt{q}}{\sqrt{\beta}m_F(x, y)}\right)^q$$

for universal constants $c_1, c_2$. We additionally have $\ell_{0\text{-}1}(F(x), y) \leq \ell_\beta(m_F(x, y))$. Because of Lemma A.3, the conditions of Lemma A.2 are satisfied, and applying Lemma A.2 with smooth loss $\ell_\beta$ gives with probability $1 - \delta$, for all $F \in \mathcal{F}$ with training error 0

$$\mathbb{E}_P[\ell_{0\text{-}1}(F(x), y)] \lesssim \tilde{E}_F(\beta) + \frac{\log(1/\delta) + \log\log n}{n}$$

where $\tilde{E}_F(\beta)$ is defined by

$$\tilde{E}_F(\beta) \triangleq \frac{1}{n} \sum_{(x,y) \in P_n} \left(\frac{c_2\sqrt{q}}{\sqrt{\beta}m_F(x, y)}\right)^q + \frac{\beta \mathcal{C}_{\|\cdot\|}^2(\mathcal{F})\log^2 n}{n}$$

and $\mathcal{C}_{\|\cdot\|}^2(\mathcal{F})$ is defined as in Lemma A.3. Choosing $\beta$ to minimize the above expression would give the desired bound – however, such a post-hoc analysis cannot be performed because the optimal $\beta$ depends on the training data, and the loss class has to be fixed before the training data is drawn.

Instead, we utilize the standard technique of union bounding over a grid of $\widehat{\beta}$ in log-scale. Let $\xi \triangleq \mathcal{C}_{\|\cdot\|}^2(\mathcal{F})\text{poly}(n^{-1})$ denote the minimum choice of $\widehat{\beta}$ in this grid, and select in this grid all choices of $\widehat{\beta}$ in the form $\xi 2^j$ for $j \geq 0$. For a given choice of $\widehat{\beta}$, we assign it failure probability $\widehat{\delta} = \frac{\delta}{2\widehat{\beta}/\xi}$, such that by design $\sum \widehat{\delta} = \delta$. Thus, applying Lemma A.2 for each choice of $\widehat{\beta}$ with corresponding failure probability $\widehat{\delta}$, we note with probability $1 - \delta$,

$$\mathbb{E}_P[\ell_{0\text{-}1}(F(x), y)] \lesssim \tilde{E}_F(\widehat{\beta}) + \frac{\log(1/\delta) + \log(\widehat{\beta}/\xi) + \log\log n}{n}$$

holds for all $\widehat{\beta}$ and $F \in \mathcal{F}$.

Now for fixed $F \in \mathcal{F}$, let $\beta_F^\star$ denote the optimizer of $\tilde{E}_F(\beta)$. We claim either there is some choice of $\widehat{\beta}$ with

$$\tilde{E}_F(\widehat{\beta}) + \frac{\log(1/\delta) + \log(\widehat{\beta}/\xi) + \log\log n}{n} \lesssim \tilde{E}_F(\beta_F^\star) + O\left(\frac{\log n + \log(1/\delta)}{n}\right) \quad \text{(A.4)}$$

or $\tilde{E}_F(\beta_F^\star) \gtrsim 1$, in which case the generalization guarantees of Theorem A.1 for this $F$ anyways trivially hold. To see this, we note that there is $\widehat{\beta}$ in the grid such that $\widehat{\beta} \in [\beta_F^\star, 2\beta_F^\star + \xi]$. Then

$$\tilde{E}_F(\widehat{\beta}) \leq \frac{1}{n} \sum_{(x,y) \in P_n} \left(\frac{c_2\sqrt{q}}{\sqrt{\beta_F^\star}m_F(x, y)}\right)^q + \frac{4\beta_F^\star \mathcal{C}_{\|\cdot\|}^2(\mathcal{F})\log^2 n}{n} + \text{poly}(n^{-1})$$

$$\leq 4\tilde{E}_F(\beta_F^\star) + \text{poly}(n^{-1})$$

Furthermore, we note that if $\beta_F^\star > \text{poly}(n)\xi$, then $\tilde{E}_F(\beta_F^\star) \gtrsim 1$. This allows to only consider $\widehat{\beta} < \text{poly}(n)\xi$, giving (A.4).

Thus, with probability $1 - \delta$, for all $F \in \mathcal{F}$, we have

$$\mathbb{E}_P[\ell_{0\text{-}1}(F(x), y)] \lesssim \tilde{E}_F(\beta_F^\star) + O\left(\frac{\log n + \log(1/\delta)}{n}\right)$$

It remains to observe that setting $\beta_F^\star = \Theta\left(q\left(\frac{n\left\|\frac{1}{m_F}\right\|_{L_q(P_n)}^q}{\mathcal{C}_{\|\cdot\|}(\mathcal{F})^2\log^2 n}\right)^{2/(q+2)}\right)$ gives us the theorem statement.

$\square$

**Claim A.2.** *For $\beta > 0$, define the loss function $\ell_\beta(m) \triangleq 1 + 2\min\{0, \Pr_{Z\sim\mathcal{N}(0,1)}(Z/\sqrt{\beta} \geq m) - 0.5\}$. Then $\ell_\beta$ satisfies the following properties:*

1. *For all $m \in \mathbb{R}$, $\ell_\beta(m) \in [0, 1]$, and for $m \leq 0$, $\ell_\beta(m) = 1$.*

2. *The function $\ell_\beta$ is $c_1\beta$-smooth for some constant $c_1$ independent of $\beta$.*

3. *For any integer $q > 0$ and $m > 0$, $\ell_\beta(m) \leq \frac{q^{q/2}c_2^q}{\beta^{q/2}m^q}$ for some constant $c_2$ independent of $q$.*

*Proof.* The first property follows directly from the construction of $\ell_\beta$. For the second property, we first note that

$$\frac{d^2}{dm^2}\Pr_{Z\sim\mathcal{N}(0,1)}(Z/\sqrt{\beta} \geq m) = \frac{m\beta^{3/2}}{\sqrt{2\pi}}\exp(-m^2\beta/2)$$

Now first note that at $m = 0$, the above quantity evaluates to 0, and thus $\ell_\beta$ has a second derivative everywhere (as $m = 0$ is the only point where the function switches). Furthermore, $\max_m m\sqrt{\beta}\exp(-m^2\beta/2) = \max_y y\exp(-y^2/2) \leq c'$ for some constant $c'$ independent of $\beta$. Thus, the above expression is upper bounded by $\frac{\beta}{\sqrt{2\pi}}c'$, giving the second property.

For the third property, we note that for $m > 0$, $\ell_\beta(m) = 2\Pr_{Z\sim\mathcal{N}(0,1)}(Z/\sqrt{\beta} \geq m)$. As the $q$-th moment of a Gaussian random variable with variance 1 is upper bounded by $q^{q/2}c_2^q$ for all $q$ and some $c_2$ independent of $q$, Markov's inequality gives the desired result. $\quad\square$

### A.1 PROOF OF LEMMA A.2

The proof is a straightforward application of Lemma A.1 and conversion of (Srebro et al., 2010) from the language of Rademacher complexity to covering numbers.

*Proof of Lemma A.2.* We can follow the proof of Theorem 1 in (Srebro et al., 2010), with the only difference that we replace their Rademacher complexity term with our complexity function $\mathcal{C}_{\|\cdot\|}(\mathcal{F})$. For completeness, we outline the steps here.

Define $\mathcal{H}(\mu) = \{h \in \ell \circ m \circ \mathcal{F} : \mathbb{E}_{P_n}[h] \leq \mu\}$ to be the class of functions in $\ell \circ m \circ \mathcal{F}$ with empirical loss at most $\mu$. Define $\psi(\mu) \triangleq \frac{\mathcal{C}_{\|\cdot\|}(\mathcal{F})\sqrt{48\beta\mu}}{\sqrt{n}}\log n$. By Claim A.3, the following holds for all $\mu$:

$$\mathbb{E}_\sigma\left[\sup_{h\in\mathcal{H}(\mu)}\sum_i \sigma_i h(x_i, y_i)\right] \leq \psi(\mu)$$

Now using the same steps as (Srebro et al., 2010) (which relies on applying Theorem 6.1 of (Bousquet, 2002)), we obtain for all $F \in \mathcal{F}$, with probability $1 - \delta$

$$\mathbb{E}_P[\ell \circ m_F] \leq \mathbb{E}_{P_n}[\ell \circ m_F] + 106r_n^\star$$
$$+ \frac{48}{n}(\log 1/\delta + \log\log n) + \sqrt{\mathbb{E}_{P_n}[\ell \circ m_F](8r_n^\star + \frac{4}{n}(\log 1/\delta + \log\log n))} \quad \text{(A.5)}$$

where $r_n^\star$ is the largest solution of $\psi(\mu) = \mu$. We now plug in $r_n^\star \lesssim \frac{\beta\log^2 n\mathcal{C}_{\|\cdot\|}^2(\mathcal{F})}{n}$ and use the fact that $\sqrt{c_1 c_2} \leq (c_1 + c_2)/2$ for any $c_1, c_2 > 0$ to simplify the square root term in A.5.

$$\mathbb{E}_P[\ell \circ m_F] \leq \frac{3}{2}\mathbb{E}_{P_n}[\ell \circ m_F] + c\left(\frac{\beta\mathcal{C}_{\|\cdot\|}^2(\mathcal{F})\log^2 n}{n} + \frac{\log(1/\delta) + \log\log n}{n}\right)$$

for some universal constant $c$. $\quad\square$

**Claim A.3.** *In the setting above, for all $\mu > 0$, we have*

$$\mathbb{E}_\sigma\left[\sup_{h\in\mathcal{H}(\mu)}\sum_i \sigma_i h(x_i, y_i)\right] \leq \frac{\mathcal{C}_{\|\cdot\|}(\mathcal{F})\sqrt{48\beta\mu}}{\sqrt{n}}\log n$$

*where $\{\sigma_i\}_{i=1}^n$ are i.i.d. Rademacher random variables.*

*Proof.* First, by Dudley's Theorem (Dudley, 1967), we have

$$\mathbb{E}_\sigma \left[ \sup_{h \in \mathcal{H}(\mu)} \sum_i \sigma_i h(x_i, y_i) \right] \leq \inf_{\alpha > 0} \left( \alpha + \frac{1}{\sqrt{n}} \int_\alpha^\infty \sqrt{\log \mathcal{N}_{L_2(P_n)}(\epsilon, \mathcal{H}(\mu))} d\epsilon \right)$$

Now by Claim A.4, we obtain

$$\mathbb{E}_\sigma \left[ \sup_{h \in \mathcal{H}(\mu)} \sum_i \sigma_i h(x_i, y_i) \right] \leq \inf_{\alpha > 0} \left( \alpha + \frac{1}{\sqrt{n}} \int_\alpha^\infty \sqrt{\left\lfloor 48\beta\mu \frac{\mathcal{C}^2_{\|\cdot\|}(\mathcal{F})}{\epsilon^2} \right\rfloor} d\epsilon \right) \quad \text{(by Claim A.4)}$$

$$\leq \inf_{\alpha > 0} \left( \alpha + \frac{\sqrt{48\beta\mu}}{\sqrt{n}} \int_{\alpha/\sqrt{48\beta\mu}}^\infty \sqrt{\left\lfloor \frac{\mathcal{C}^2_{\|\cdot\|}(\mathcal{F})}{\epsilon'^2} \right\rfloor} d\epsilon' \right)$$

$$(\text{A.6})$$

We obtained the last line via change of variables to $\epsilon' = \epsilon/\sqrt{48\beta\mu}$. Now we substitute $\alpha = \frac{\mathcal{C}_{\|\cdot\|}(\mathcal{F})\sqrt{48\beta\mu}}{\sqrt{n}}$ and note that the integrand is 0 for $\epsilon' > \mathcal{C}_{\|\cdot\|}(\mathcal{F})$ to get

$$\mathbb{E}_\sigma \left[ \sup_{h \in \mathcal{H}(\mu)} \sum_i \sigma_i h(x_i, y_i) \right] \leq \frac{\mathcal{C}_{\|\cdot\|}(\mathcal{F})\sqrt{48\beta\mu}}{\sqrt{n}} \left( 1 + \int_{\mathcal{C}_{\|\cdot\|}(\mathcal{F})/\sqrt{n}}^{\mathcal{C}_{\|\cdot\|}(\mathcal{F})} \frac{1}{\epsilon'} d\epsilon' \right)$$

$$\leq \frac{\mathcal{C}_{\|\cdot\|}(\mathcal{F})\sqrt{48\beta\mu}}{\sqrt{n}} \log n$$

$$\square$$

The following claim applies Lemma A.1 in order to bound the covering number of $\mathcal{H}(\mu)$ in terms of $\mathcal{C}_{\|\cdot\|}(\mathcal{F})$.

**Claim A.4.** *In the setting of Lemma A.2, we have the covering number bound*

$$\log \mathcal{N}_{L_2(P_n)}(\epsilon, \mathcal{H}(\mu)) \leq \left\lfloor 48\beta\mu \frac{\mathcal{C}_{\|\cdot\|}(\mathcal{F})}{\epsilon^2} \right\rfloor$$

*Proof.* As $\ell \circ m \circ \mathcal{F}$ is the composition of a $\beta$-smooth loss $\ell$ with the function class $m \circ \mathcal{F}$, by equation (22) of (Srebro et al., 2010) we have

$$\log \mathcal{N}_{L_2(P_n)}(\epsilon, \mathcal{H}(\mu)) \leq \log \mathcal{N}_\infty(\epsilon/\sqrt{48\beta\mu}, m \circ \mathcal{F})$$

$$\leq \log \mathcal{N}_{\|\cdot\|}(\epsilon/\sqrt{48\beta\mu}, \mathcal{F}) \quad \text{(by Lemma A.1)}$$

$$\leq \left\lfloor 48\beta\mu \frac{\mathcal{C}_{\|\cdot\|}(\mathcal{F})}{\epsilon^2} \right\rfloor \quad \text{(as } \mathcal{F} \text{ satisfies Condition A.1)}$$

$$\square$$

## B  PROOFS FOR NEURAL NET GENERALIZATION

This section will derive the generalization bounds for neural nets in Theorem 3.1 by invoking the more general results in Section C. Theorem 3.1 applies to all neural nets, but to obtain it, we first need to bound generalization for neural nets with *fixed* norm bounds on their weights (this is a standard step in deriving generalization bounds). The lemma below states the analogue of Theorem 3.1, for all neural nets satisfying fixed norm bounds on their weights.

**Lemma B.1.** *In the neural network setting, suppose that the activation $\phi$ has a $\kappa'_\phi$-Lipschitz derivative. For parameters $\{a_{(i)}\}_{i=1}^r$ meant to be norm constraints for the weights, define the class of neural nets with bounded weight norms with respect to reference matrices $\{A_{(i)}, B_{(i)}\}$ as follows:*

$$\mathcal{F} \triangleq \{x \mapsto F(x) : \min\{\sqrt{d}\|W_{(i)} - A_{(i)}\|_{\text{fro}}, \|W_{(i)} - B_{(i)}\|_{1,1}\} \sqrt{\log d} \leq a_{(i)} \; \forall i\}$$

*Then with probability $1 - \delta$, for any $q > 0$ and for all $F \in \mathcal{F}$, we have*

$$\mathbb{E}_P[\ell_{0\text{-}1}(F(x), y)]$$

$$\leq \frac{3}{2}\mathbb{E}_{P_n}[\ell_{0\text{-}1}(F(x), y)] + O\left(\frac{r\log n + \log(1/\delta)}{n}\right)$$

$$+ (1 - \mathbb{E}_{P_n}[\ell_{0\text{-}1}(F(x), y)])^{2/(q+2)} O\left(q\left(\frac{\log^2 n}{n}\right)^{\frac{q}{(q+2)}}\left(\sum_i (\|\kappa^{\text{NN}}_{(i)}\|_{L_q(\mathcal{S}_n)} a_{(i)})^{2/3}\right)^{\frac{3q}{(q+2)}}\right)$$

*where $\mathcal{S}_n$ denotes the subset of examples classified correctly by $F$ and $\kappa^{\text{NN}}_{(i)}$ is defined as in* (3.2).

*Proof.* We will identify the class of neural nets with matrix norm bounds $\{a_{(i)}\}_{i=1}^r$ with a sequence of function families

$$\mathcal{F}_{2i-1} \triangleq \{h \mapsto Wh : W \in \mathbb{R}^{d\times d}, \min\{\sqrt{d}\|W - A_{(i)}\|_F, \|W - B_{(i)}\|_{1,1}\}\sqrt{\log d} \leq a_{(i)}\}$$

$$\mathcal{F}_{2i} \triangleq \{h \mapsto \phi(h)\}$$

and let $\mathcal{F} \triangleq \mathcal{F}_{2r-1} \circ \cdots \circ \mathcal{F}_1$ denote all possible parameterizations of neural nets with norm bounds $\{a_{(i)}\}_{i=1}^r$. Let $\|\cdot\|_{\text{op}}$ be defined with respect to Euclidean norm $\|\cdot\|_2$ on the input and output spaces, which coincides with matrix operator norm for linear operators. We first claim that

$$\log\mathcal{N}_{\|\cdot\|_{\text{op}}}(\epsilon, \mathcal{F}_{2i-1}) \lesssim \left\lfloor \frac{a_{(i)}^2}{\epsilon^2} \right\rfloor$$

This is because we can construct two covers: one for $\{h \mapsto Wh : \sqrt{d}\|W\|_F/\sqrt{\log d} \leq a_{(i)}\}$, and one for $\{h \mapsto Wh : \|W\|_{1,1}/\sqrt{\log d} \leq a_{(i)}\}$, each of which has log size bounded by $O(\lfloor a_{(i)}^2/\epsilon^2 \rfloor)$ by Lemma B.2 and Claim B.2. Now we offset the first cover by the linear operator $A_{(i)}$ and the second by $B_{(i)}$ and take the union of the two, obtaining an $\epsilon$-cover for $\mathcal{F}_{2i-1}$ in operator norm. Furthermore, $\log\mathcal{N}_{\|\cdot\|_{\text{op}}}(\epsilon, \mathcal{F}_{2i}) = 0$ simply because $\mathcal{F}_{2i}$ is the singleton function.

Thus, $\mathcal{F}_{2i-1}, \mathcal{F}_{2i}$ satisfy Condition A.1 with norm $\|\cdot\|_{\text{op}}$ and complexity functions $\mathcal{C}_{\|\cdot\|_{\text{op}}}(\mathcal{F}_{2i-1}) \lesssim a_{(i)}$ and $\mathcal{C}_{\|\cdot\|_{\text{op}}}(\mathcal{F}_{2i}) = 0$, so we can apply Theorem C.1. It remains to argue that $\kappa^{\star}_{2i-1}(x, y)$ as defined for Theorem C.1 using standard Euclidean norm $\|\cdot\|_2$ is equivalent to $\kappa^{\text{NN}}_{(i)}(x, y)$ defined in (3.2). To see this, we note that functions in $\mathcal{F}_{2j-1}$ have 0-Lipschitz derivative, leading those terms with a coefficient of $\kappa'_{2j-1}$ to cancel in the definition of $\kappa^{\star}_i(x, y)$. There is a 1-1 correspondence between the remaining terms of $\kappa^{\star}_{2i-1}(x, y)$ and $\kappa^{\text{NN}}_{(i)}(x, y)$, so we can substitute $\kappa^{\text{NN}}_{(i)}(x, y)$ into Theorem C.1 in place of $\kappa^{\star}_{2i-1}(x, y)$. Furthermore, as we have $\mathcal{C}_{\|\cdot\|_{\text{op}}}(\mathcal{F}_{2i}) = 0$, the corresponding terms disappear in the bound of Theorem C.1, finally giving the desired result.

$\square$

Now we obtain Theorem 3.1 by union bounding Lemma B.1 over choices of $\{a_{(i)}\}_{i=1}^r$.

*Proof of Theorem 3.1.* We will use the standard technique of applying Lemma B.1 over many choices of $\{a_{(i)}\}$, and union bounding over the failure probability. Choose $\xi = \text{poly}(n^{-1})$ and consider a grid of $\{\widehat{\alpha}_{(i)}\}$ with $\widehat{a}_{(i)} = \xi 2^{j_i}$ for $j_i \geq 1$. We apply Lemma B.1 with for all possible norm bounds $\{\widehat{\alpha}_{(i)}\}$ in the grid, using failure probability $\widehat{\delta} = \delta/(\prod_i \widehat{a}_{(i)}/\xi)$ for a given choice of $\{\widehat{\alpha}_{(i)}\}$. By union bound, with probability $1 - \sum \widehat{\delta} = 1 - \delta$, the bound of Lemma B.1 holds simultaneously for all choices of $\{\widehat{\alpha}_{(i)}\}$. In particular, for the neural net $F$ with parameters $\{W_{(i)}\}$, there is a choice of $\{\widehat{\alpha}_{(i)}\}$ satisfying

$$\min\{\sqrt{d}\|W_{(i)} - A_{(i)}\|_{\text{fro}}, \|W_{(i)} - B_{(i)}\|_{1,1}\}\sqrt{\log d}$$

$$\leq \widehat{a}_{(i)} \leq 2\min\{\sqrt{d}\|W_{(i)} - A_{(i)}\|_{\text{fro}}, \|W_{(i)} - B_{(i)}\|_{1,1}\}\sqrt{\log d} + \xi$$

for all $i$. The application of Lemma B.1 for this choice of $\widehat{\alpha}_{(i)}$ gives us the desired generalization bound. $\square$

### B.1    GENERALIZATION BOUND FOR RELU NETWORKS

In the case where $\phi$ is the relu activation, we can no longer lower bound the all-layer margin $m_F(x, y)$ using the techniques in Section C, which rely on smoothness. However, we can still obtain a generalization bound in terms of the distribution of $1/m_F(x, y)$ on the training data. We can expect $1/m_F(x, y)$ to be small in practice because relu networks typically exhibit stability to perturbations. Prior bounds for relu nets suffer from some source of looseness: the bounds of (Bartlett et al., 2017; Neyshabur et al., 2017b) depended on the product of weight norms divided by margin, and the bounds of (Nagarajan and Kolter, 2019) depended on the inverse of the pre-activations, observed to be large in practice. Our bound avoids these dependencies, and in fact, it is possible to upper bound our dependency on $1/m_F(x, y)$ in terms of both these quantities.

For this setting, we choose a fixed $\||\cdot\||$ defined as follows: if $i$ corresponds to a linear layer in the network, set $\alpha_i = 1$, and for $i$ corresponding to activation layers, set $\alpha_i = \infty$ (in other words, we only allow perturbations after linear layers). We remark that we could use alternative definitions of $\||\cdot\||$, but because we do not have a closed-form lower bound on $m_F$, the tradeoff between these formulations is unclear.

**Theorem B.1.** *In the neural network setting, suppose that $\phi$ is any activation (such as the relu function) and $m_F$ is defined using $\||\cdot\||$ as described above. Fix any integer $q > 0$. Then with probability $1 - \delta$, for all relu networks $F$ parameterized by weight matrices $\{W_{(i)}\}_{i=1}^r$ that achieve training error 0, we have*

$$\mathbb{E}_P[\ell_{\text{0-1}} \circ F] \leq O\left(\log^2 n \left(\frac{\left\|\frac{1}{m_F}\right\|_{L_q(P_n)}\left(\sum_i a_{(i)}\right)}{\sqrt{n}}\right)^{2q/(q+2)}\right) + \zeta$$

*where $a_{(i)}$ is defined as in Theorem 3.1, and $\zeta \triangleq O\left(\frac{\log(1/\delta) + r\log n + \sum_i(a_{(i)}+1)}{n}\right)$ is a low-order term.*

The proof follows via direct application of Theorem A.1 and the same arguments as Lemma B.1 relating matrix norms to covering numbers. We remark that in the case of relu networks, we can upper bound $\frac{1}{m_F(x,y)}$ via a quantity depending on the inverse pre-activations that mirrors the bound of Nagarajan and Kolter (2019). However, as mentioned earlier, this is a pessimistic upper bound as Nagarajan and Kolter (2019) show that the inverse preactivations can be quite large in practice.

### B.2    MATRIX COVERING LEMMAS

In this section we present our spectral norm cover for the weight matrices, which is used in Section B to prove our neural net generalization bounds.

**Lemma B.2.** *Let $\mathcal{M}_{\text{fro}}(B)$ denote the set of $d_1 \times d_2$ matrices with Frobenius norm bounded by $B$, i.e.*

$$\mathcal{M}_{\text{fro}}(B) \triangleq \{M \in \mathbb{R}^{d_1 \times d_2} : \|M\|_{\text{fro}} \leq B\}$$

*Then letting $d \triangleq \max\{d_1, d_2\}$ denote the larger dimension, for all $\epsilon > 0$, we have*

$$\log \mathcal{N}_{\|\cdot\|_{\text{op}}}(\epsilon, \mathcal{M}_{\text{fro}}(B)) \leq \left\lfloor \frac{36dB^2\log(9d)}{\epsilon^2} \right\rfloor$$

*Proof.* The idea for this proof is that since the cover is in spectral norm, we only need to cover the top $d' \triangleq \lfloor B^2/\epsilon^2 \rfloor$ singular vectors of matrices $M \in \mathcal{M}$.

First, it suffices to work with square matrices, as a spectral norm cover of $\max\{d_1, d_2\} \times \max\{d_1, d_2\}$ matrices will also yield a cover of $d_1 \times d_2$ matrices in spectral norm (as we can extend a $d_1 \times d_2$ matrices to a larger square matrix by adding rows or columns with all 0). Thus, letting $d \triangleq \max\{d_1, d_2\}$, we will cover $\mathcal{M}_{\text{fro}}(B)$ defined with respect to $d \times d$ matrices.

Let $d' \triangleq \lfloor 9B^2/\epsilon^2 \rfloor$. We first work in the case when $d' \leq d$. Let $\widehat{\mathcal{U}}$ be a $\epsilon_U$ Frobenius norm cover of $d \times d'$ matrices with Frobenius norm bound $d'$. Let $\widehat{\mathcal{V}}$ be the cover of $d' \times d$ matrices with Frobenius

norm bound $B$ in Frobenius norm with resolution $\epsilon_V$. We construct a cover $\widehat{\mathcal{M}}$ for $\mathcal{M}_{\mathrm{fro}}(B)$ as follows: take all possible combinations of matrices $\widehat{U}, \widehat{V}$ from $\widehat{\mathcal{U}}, \widehat{\mathcal{V}}$, and add $\widehat{U}\widehat{V}$ to $\widehat{\mathcal{M}}$. First note that by Claim B.1, we have

$$\log |\widehat{\mathcal{M}}| \leq dd'(\log(3d'/\epsilon_U) + \log(3B/\epsilon_V))$$

Now we analyze the cover resolution of $\widehat{\mathcal{M}}$: for $M \in \mathcal{M}$, first let $\mathrm{trunc}_{d'}(M)$ be the truncation of $M$ to its $d'$ largest singular values. Note that as $M$ has at most $d'$ singular values with absolute value greater than $\epsilon/3$, $\|M - \mathrm{trunc}_{d'}(M)\|_{\mathrm{op}} \leq \epsilon/3$. Furthermore, let $USV = \mathrm{trunc}_{d'}(M)$ be the SVD decomposition of this truncation, where $U \in \mathbb{R}^{d \times d'}$, $\|U\|_{\mathrm{fro}} \leq d'$ and $SV \in \mathbb{R}^{d' \times d}$, $\|SV\|_{\mathrm{fro}} \leq B$. Let $\widehat{U} \in \widehat{\mathcal{U}}$ satisfy $\|\widehat{U} - U\|_{\mathrm{fro}} \leq \epsilon_U$, and $\widehat{V} \in \widehat{\mathcal{V}}$ satisfy $\|\widehat{V} - SV\|_{\mathrm{fro}} \leq \epsilon_V$. Let $\widehat{M} = \widehat{U}\widehat{V}$. Then we obtain

$$
\begin{aligned}
\|M - \widehat{M}\|_{\mathrm{op}} &\leq \|M - \mathrm{trunc}_{d'}(M)\|_{\mathrm{op}} + \|\mathrm{trunc}_{d'}(M) - \widehat{M}\|_{\mathrm{op}} \\
&\leq \epsilon/3 + \|USV - \widehat{U}SV\|_{\mathrm{op}} + \|\widehat{U}SV - \widehat{U}\widehat{V}\|_{\mathrm{op}} \\
&\leq \epsilon + \epsilon_U B + \epsilon_V d'
\end{aligned}
$$

Thus, setting $\epsilon_U = \epsilon/3B$, $\epsilon_V = \epsilon/3d'$, then we get a $\epsilon$-cover of $\mathcal{M}$ with log cover size $\lfloor 9dB^2/\epsilon^2 \rfloor (\log 81 d'^2 B^2/\epsilon^2)$. As $d' \leq d$, this simplifies to $\lfloor 36dB^2 \log(9d)/\epsilon^2 \rfloor$.

Now when $d' \geq d$, we simply take a Frobenius norm cover of $d \times d$ matrices with Frobenius norm bound $B$, which by Claim B.1 has log size at most $d^2 \log(3B/\epsilon) \leq \lfloor 36dB^2 \log(9d)/\epsilon^2 \rfloor$, where the inequality followed because $9B^2/\epsilon^2 \geq d$.

Combining both cases, we get for all $\epsilon > 0$,

$$\log \mathcal{N}_{\|\cdot\|_{\mathrm{op}}}(\epsilon, \mathcal{M}_{\mathrm{fro}}(B)) \leq \left\lfloor \frac{36dB^2 \log(9d)}{\epsilon^2} \right\rfloor$$

$\square$

The following claims are straightforward and follow from standard covering number bounds for $\|\cdot\|_2$ and $\|\cdot\|_1$ balls.

**Claim B.1.** *Let $\mathcal{M}_{\mathrm{fro}}(B)$ denote the class of $d_1 \times d_2$ matrices with Frobenius norm bounded by $B$. Then for $0 < \epsilon < B$, $\log \mathcal{N}_{\|\cdot\|_{\mathrm{fro}}}(\epsilon, \mathcal{M}_{\mathrm{fro}}(B)) \leq d_1 d_2 \log(3B/\epsilon)$.*

**Claim B.2.** *Let $\mathcal{M}_{\|\cdot\|_{1,1}}(B)$ denote the class of $d_1 \times d_2$ matrices with the $\ell_1$ norm of its entries bounded by $B$. Then $\log \mathcal{N}_{\|\cdot\|_{\mathrm{fro}}}(\epsilon, \mathcal{M}_{\|\cdot\|_{1,1}}(B)) \leq 5 \lfloor B^2/\epsilon^2 \rfloor \log 10d$.*

## C  GENERALIZATION BOUND FOR SMOOTH FUNCTION COMPOSITIONS

In this section, we present the bound for general smooth function compositions used to prove Theorem 3.1.

We will work in the same general setting as Section A. Let $J_{j \leftarrow i}(x, \delta)$ denote the $i$-to-$j$ Jacobian evaluated at $f_{i-1 \leftarrow 1}(x, \delta)$, i.e. $J_{j \leftarrow i}(x, \delta) \triangleq D_h f_{j \leftarrow i}(h, \delta)|_{h = f_{i-1 \leftarrow 1}(x, \delta)}$. We will additionally define general notation for hidden layer and Jacobian norms which coincides with our notation for neural nets. Let $s_i(x) \triangleq \|f_{i \leftarrow 1}(x)\|$ and $s_0(x) \triangleq \|x\|$. As the function $Df_{j \leftarrow i}$ outputs operators mapping $\mathcal{D}_{i-1}$ to $\mathcal{D}_j$, we can additionally define $\kappa_{j \leftarrow i}(x) \triangleq \|Df_{j \leftarrow i} \circ f_{i-1 \leftarrow 1}(x)\|_{\mathrm{op}}$, with $\kappa_{j \leftarrow j+1}(x) \triangleq 1$.

Let $\kappa_i'$ be an upper bound on the Lipschitz constant of $Df_{i \leftarrow i}$ measured in operator norm:

$$\|Df_{i \leftarrow i}(h) - Df_{i \leftarrow i}(h + \nu)\|_{\mathrm{op}} \leq \kappa_i' \|\nu\|$$

Now we define the value $\kappa_i^\star(x, y)$, which can be thought of as a Lipschitz constant for perturbation $\delta_i$ in the definition of $m_F$, as follows:

$$
\begin{aligned}
\kappa_i^\star(x, y) \triangleq\ & s_{i-1}(x) \left( \frac{8\kappa_{k \leftarrow i+1}(x)}{\gamma(F(x), y)} + \sum_{j=i}^{k-1} \frac{8\kappa_{j \leftarrow i+1}(x)}{s_j(x)} \right) \\
& + s_{i-1}(x) \left( \sum_{1 \leq j_2 \leq j_1 \leq k} \sum_{j'=\max\{i+1, j_2\}}^{j_1} 16 \frac{\kappa'_{j'} \kappa_{j'-1 \leftarrow i+1}(x) \kappa_{j_1 \leftarrow j'+1}(x) \kappa_{j'-1 \leftarrow j_2}(x)}{\kappa_{j_1 \leftarrow j_2}(x)} \right) \\
& + 8 \sum_{j_2 \leq i \leq j_1} \frac{\kappa_{j_1 \leftarrow i+1}(x) \kappa_{i-1 \leftarrow j_2}(x)}{\kappa_{j_1 \leftarrow j_2}(x)}
\end{aligned}
\tag{C.1}
$$

For this general setting, the following theorem implies that for any integer $q > 0$, if $F$ classifies all training examples correctly, then its error converges at a rate that scales with $n^{-q/(q+2)}$ and the products $\|\kappa_i^\star\|_{L_q(P_n)} \mathcal{C}_{\|\cdot\|_{\mathrm{op}}}(\mathcal{F}_i)$.

**Theorem C.1.** *Let $\mathcal{F} = \{f_k \circ \cdots \circ f_1 : f_i \in \mathcal{F}_i\}$ denote a class of compositions of functions from $k$ families $\{\mathcal{F}_i\}_{i=1}^k$, each of which satisfies Condition A.1 with operator norm $\|\cdot\|_{\mathrm{op}}$ and complexity $\mathcal{C}_{\|\cdot\|_{\mathrm{op}}}(\mathcal{F}_i)$. For any choice of integer $q > 0$, with probability $1 - \delta$ for all $F \in \mathcal{F}$ the following bound holds:*

$$
\mathbb{E}_P[\ell_{0\text{-}1}(F(x), y)]
$$
$$
\leq \frac{3}{2} \left( \mathbb{E}_{P_n}[\ell_{0\text{-}1}(F(x), y)] \right) + O\left( \frac{k \log n + \log(1/\delta)}{n} \right)
$$
$$
+ (1 - \mathbb{E}_{P_n}[\ell_{0\text{-}1}(F(x), y)])^{\frac{2}{q+2}} O\left( q \left( \frac{\log^2 n}{n} \right)^{q/(q+2)} \left( \sum_i \|\kappa_i^\star\|_{L_q(\mathcal{S}_n)}^{2/3} \mathcal{C}_{\|\cdot\|_{\mathrm{op}}}(\mathcal{F}_i)^{2/3} \right)^{\frac{3q}{q+2}} \right)
$$

*where $\mathcal{S}_n$ denotes the subset of training examples correctly classified by $F$ and $\kappa_i^\star$ is defined in (C.1). In particular, if $F$ classifies all training samples correctly, i.e. $|\mathcal{S}_n| = n$, with probability $1 - \delta$ we have*

$$
\mathbb{E}_P[\ell_{0\text{-}1}(F(x), y)] \lesssim q \left( \frac{\log^2 n}{n} \right)^{q/(q+2)} \left( \sum_i \|\kappa_i^\star\|_{L_q(\mathcal{S}_n)}^{2/3} \mathcal{C}_{\|\cdot\|_{\mathrm{op}}}(\mathcal{F}_i)^{2/3} \right)^{3q/(q+2)} + \zeta
$$

*where $\zeta \triangleq O\left( \frac{k \log n + \log(1/\delta)}{n} \right)$ is a low order term.*

To prove this theorem, we will plug the following lower bound on $m_F$ into Lemma A.2 with the appropriate choice of smooth loss $\ell$, and pick the optimal choice of $\{\alpha_i\}_{i=1}^k$ for the resulting bound. We remark that we could also use Theorem A.1 as our starting point, but this would still require optimizing over $\{\alpha_i\}_{i=1}^k$.

**Lemma C.1** (General version of Lemma 3.1). *In the setting of Theorem C.1, where each layer $\mathcal{F}_i$ is a class of smooth functions, if $\gamma(F(x), y) > 0$, we have*

$$
m_F(x, y) \geq \|\{\kappa_i^\star(x, y)/\alpha_i\}_{i=1}^k\|_{p/(p-1)}^{-1}
$$

We prove Lemma C.1 in Section D by formalizing the intuition outlined in Section 3. With Lemma C.1 in hand, we can prove Theorem C.1. This proof will follow the same outline as the proof of Theorem A.1. The primary difference is that we optimize over $k$ values of $\alpha_i$, whereas Theorem A.1 only optimized over the smoothness $\beta$.

*Proof of Theorem C.1.* We use $\ell_\beta$ with $\beta = 1$ defined in Claim A.2 as a surrogate loss for the 0-1 loss. Since Claim A.2 gives $\ell_{0\text{-}1}(F(x), y) \leq \ell_{\beta=1}(m_F(x, y))$, by Lemma A.2 it follows that

$$
\mathbb{E}_P[\ell_{0\text{-}1}(F(x), y)] \leq \mathbb{E}_P[\ell_{\beta=1}(m_F(x, y))]
$$
$$
\leq \frac{3}{2} \mathbb{E}_{P_n}[\ell_{\beta=1}(m_F(x, y))] + c_1 \left( \frac{\mathcal{C}_{\|\cdot\|}^2(\mathcal{F}) \log^2 n}{n} + \frac{\log(1/\delta) + \log \log n}{n} \right)
\tag{C.2}
$$

Now we first note that for a misclassified pair, $\ell_{\beta=1}(m_F(x,y)) = \ell_{0\text{-}1}(F(x),y) = 1$. For correctly classified examples, we also have the bound $\ell_{\beta=1}(m_F(x,y)) \leq \frac{(c_2 q)^{q/2}}{m_F(x,y)^q}$ for constant $c_2$ independent of $q$. Thus, it follows that

$$\mathbb{E}_{P_n}[\ell_{\beta\text{-}1}(m_F(x,y))] \leq \mathbb{E}_{P_n}[\ell_{0\text{-}1}(F(x),y)] + \frac{1}{n} \sum_{(x,y) \in \mathcal{S}_n} \frac{(c_2 q)^{q/2}}{m_F(x,y)^q}$$

Plugging this into (C.2), we get with probability $1 - \delta$ for all $F \in \mathcal{F}$,

$$\mathbb{E}_{P_n}[\ell_{\beta\text{-}1}(m_F(x,y))] \leq \frac{3}{2}\mathbb{E}_{P_n}[\ell_{0\text{-}1}(F(x),y)] + O\left(E + \frac{\log(1/\delta) + \log\log n}{n}\right) \qquad \text{(C.3)}$$

where $E$ is defined by

$$E = \frac{1}{n} \sum_{(x,y) \in \mathcal{S}_n} \frac{(c_2 q)^{q/2}}{m_F(x,y)^q} + \frac{\mathcal{C}_{\|\cdot\|}^2(\mathcal{F})\log^2 n}{n} \qquad \text{(C.4)}$$

Thus, it suffices to upper bound $E$. By Lemma C.1, we have $m_F(x,y) \geq \|\{\kappa_i^\star(x,y)/\alpha_i\}_{i=1}^k\|_{p/(p-1)}^{-1}$ for the choice of $\alpha, p$ used to define $m_F$. We will set $p = q/(q-1)$ and union bound (C.3) over choices of $\alpha$.

First, for a particular choice of $\alpha$ and $p = q/(q-1)$, we apply our lower bound on $m_F(x,y)$ to simplify (C.4) as follows:

$$E \leq \frac{1}{n} \sum_{(x,y) \in \mathcal{S}_n} (c_2 q)^{q/2}\|(\kappa_i^\star(x,y)/\alpha_i)_{i=1}^k\|_q^q + \frac{\mathcal{C}_{\|\cdot\|}^2(\mathcal{F})\log^2 n}{n}$$

$$\leq \sum_{i=1}^k \alpha_i^{-q}\left[\frac{(c_2 q)^{q/2}}{n}\sum_{(x,y) \in \mathcal{S}_n}\kappa_i^\star(x,y)^q\right] + \left(\sum_i \alpha_i^{2q/(3q-2)}\mathcal{C}_{\|\cdot\|_{\mathrm{op}}}(\mathcal{F}_i)^{2q/(3q-2)}\right)^{\frac{3q-2}{q}}\frac{\log^2 n}{n} \qquad \text{(C.5)}$$

For convenience, we use $\tilde{E}_F(\alpha)$ to denote (C.5) as a function of $\alpha$. Note that $\kappa_i^\star$ depends on $F$. Now let $\alpha_F^\star$ denote the minimizer of $\tilde{E}_F(\alpha)$. As we do not know the exact value of $\alpha_F^\star$ before the training data is drawn, we cannot simply plug the exact value of $\alpha_F^\star$ into (C.5). Instead, we will apply a similar union bound as the proof of Theorem A.1, although this union bound is slightly more complicated because we optimize over $k$ quantities simultaneously.

We use $\xi_i$ to denote the lower limit on $\alpha_i$ in our search over $\alpha$, setting $\xi_i = \mathcal{C}_{\|\cdot\|_{\mathrm{op}}}(\mathcal{F}_i)^{-1}\text{poly}(k^{-1}n^{-1})$.[4] Now we consider a grid of $\{\widehat{\alpha}_i\}_{i=1}^k$, where $\widehat{\alpha}$ has entries of the form $\widehat{\alpha}_i = \xi_i 2^j$ for any $j \geq 0$. For a given choice of $\widehat{\alpha}$, we assign it failure probability

$$\widehat{\delta} = \frac{\delta}{\prod_i 2\widehat{\alpha}_i/\xi} \qquad \text{(C.6)}$$

where $\delta$ is the target failure probability after union bounding. First, note that

$$\sum \widehat{\delta} = \delta \sum_{j_1 \geq 0} \cdots \sum_{j_k \geq 0} \frac{1}{2^{j_1 + \cdots + j_k + k}} \leq \delta$$

Therefore, with probability $1 - \delta$, we get that (C.2) holds for $m_F$ defined with respect to every $\widehat{\alpha}$. In particular, with probability $1 - \delta$, for all $F \in \mathcal{F}$ and $\widehat{\alpha}$ in the grid,

$$\begin{aligned}
&\mathbb{E}_P[\ell_{0\text{-}1}(F(x),y)] \\
&\leq \mathbb{E}_{P_n}[\ell_{\beta=1}(m_F(x,y))] \\
&\leq \frac{3}{2}\mathbb{E}_{P_n}[\ell_{0\text{-}1}(F(x),y)] + O\left(\tilde{E}_F(\widehat{\alpha}) + \frac{\sum_i \log(2\widehat{\alpha}_i/\xi_i) + \log(1/\delta) + \log\log n}{n}\right)
\end{aligned} \qquad \text{(C.7)}$$

---

[4]If $\mathcal{C}_{\|\cdot\|_{\mathrm{op}}}(\mathcal{F}_i) = 0$, then we simply set $\alpha_i = \infty$, which is equivalent to restricting the perturbations used in computing $m_F$ to layers where $\mathcal{C}_{\|\cdot\|_{\mathrm{op}}}(\mathcal{F}_i) > 0$.

where the last term was obtained by subsituting (C.6) for the failure probability.

Now we claim that there is some choice of $\widehat{\alpha}$ in the grid such that either

$$\tilde{E}_F(\widehat{\alpha}) + \frac{\sum_i \log(2\widehat{\alpha}_i/\xi_i) + \log(1/\delta) + \log\log n}{n} \leq 9\tilde{E}_F(\alpha_F^\star) + O\left(\frac{k\log(n) + \log(1/\delta)}{n}\right) \tag{C.8}$$

or $\tilde{E}_F(\alpha_F^\star) \gtrsim 1$ (in which case it is trivial to obtain generalization error bounded by $\tilde{E}_F(\alpha_F^\star)$).

To see this, we first consider $\widehat{\alpha}$ in our grid such that $\widehat{\alpha}_i \in [\alpha_{F,i}^\star, 2\alpha_{F,i}^\star + \xi_i]$. By construction of our grid of $\widehat{\alpha}$, such a choice always exists. Then we have

$$\tilde{E}_F(\widehat{\alpha})$$

$$= \sum_{i=1}^k \widehat{\alpha}_i^{-q}\left[\frac{(c_2 q)^{q/2}}{n}\sum_{(x,y)\in\mathcal{S}_n}\kappa_i^\star(x,y)^q\right] + \left(\sum_i \widehat{\alpha}_i^{2q/(3q-2)}\mathcal{C}_{\|\cdot\|_{op}}(\mathcal{F}_i)^{2q/(3q-2)}\right)^{\frac{3q-2}{q}}\frac{\log^2 n}{n}$$

$$\leq \sum_{i=1}^k \alpha_{F,i}^{\star-q}\left[\frac{(c_2 q)^{q/2}}{n}\sum_{(x,y)\in\mathcal{S}_n}\kappa_i^\star(x,y)^q\right]$$

$$+ 9\left(\sum_i \alpha_{F,i}^{\star\,2q/(3q-2)}\mathcal{C}_{\|\cdot\|_{op}}(\mathcal{F}_i)^{2q/(3q-2)}\right)^{(3q-2)/q}\frac{\log^2 n}{n} + \text{poly}(n^{-1})$$

The first term we obtained because $\widehat{\alpha}_i \geq \alpha_{F,i}^\star$, and the second via the upper bound $\widehat{\alpha}_i \leq 2\alpha_{F,i}^\star + \xi_i$. Thus, for some choice of $\widehat{\alpha}$ in the grid, we have $\tilde{E}_F(\widehat{\alpha}) \leq 9\tilde{E}_F(\alpha_F^\star) + \text{poly}(n^{-1})$. Furthermore, if $\alpha_F^\star > c \cdot \mathcal{C}_{\|\cdot\|_{op}}(\mathcal{F}_i)^{-1}n$ for some constant $c$, we note that $\tilde{E}_F(\alpha_F^\star) \gtrsim 1$ - thus, it suffices to only consider $\alpha_F^\star \leq c \cdot \mathcal{C}_{\|\cdot\|_{op}}(\mathcal{F}_i)^{-1}n$. In particular, we only need to consider $\widehat{\alpha}_i$ where $\log(2\widehat{\alpha}_i/\xi_i) \lesssim \log kn$. Finally, we note that we can assume WLOG that $k \lesssim n$ otherwise (C.8) would give a trivial bound. Combining these facts gives (C.8).

Thus, it follows that for all $F \in \mathcal{F}$,

$$\mathbb{E}_P[\ell_{0\text{-}1}(F(x),y)] \leq \frac{3}{2}\mathbb{E}_{P_n}[\ell_{0\text{-}1}(F(x),y)] + O\left(\tilde{E}_F(\alpha_F^\star) + \frac{k\log n + \log(1/\delta)}{n}\right) \tag{C.9}$$

Finally, we can apply Lemma C.2 using $z_i = \left(\frac{(c_2 q)^{q/2}}{n}\sum_{(x,y)\in\mathcal{S}_n}\kappa_i^\star(x,y)^q\right)^{1/q} = \left(\frac{|\mathcal{S}_n|}{n}\right)^{1/q}\|\kappa_i^\star\|_{L_q(\mathcal{S}_n)}$ and $b_i = \frac{\mathcal{C}_{\|\cdot\|_{op}}(\mathcal{F}_i)\log n}{\sqrt{n}}$ to get

$$\tilde{E}_F(\alpha_F^\star) \lesssim \left(\frac{|\mathcal{S}_n|}{n}\right)^{2/(q+2)}q\left(\frac{\log^2 n}{n}\right)^{q/(q+2)}\left(\sum_i \|\kappa_i^\star\|_{L_q(\mathcal{S}_n)}^{2/3}\mathcal{C}_{\|\cdot\|_{op}}(\mathcal{F}_i)^{2/3}\right)^{3q/(q+2)}$$

Substituting into (C.9) gives the desired bound.

$\square$

**Lemma C.2.** *For coefficients $\{z_i\}_{i=1}^k, \{b_i\}_{i=1}^k > 0$ and integer $q > 0$, define*

$$E(\alpha) \triangleq \sum_i z_i^q/\alpha_i^q + \left(\sum_i \alpha_i^{2q/(3q-2)}b_i^{2q/(3q-2)}\right)^{(3q-2)/q}$$

*with minimizer $\alpha^\star$ and minimum value $E^\star$. Then*

$$E^\star \leq 2\left(\sum_i (z_i b_i)^{2/3}\right)^{3q/(q+2)}$$

*Proof.* Choose $\{\alpha_i\}_{i=1}^k$ as follows (we obtained this by solving for $\alpha$ for which $\nabla_\alpha E(\alpha) = 0$):

$$\alpha_i = \left(\frac{q}{2}\right)^{\frac{1}{(q+2)}} z_i^{\frac{3q-2}{3q}} b_i^{-\frac{2}{3q}} \left[\sum_i z_i^{2/3} b_i^{2/3}\right]^{\frac{2-2q}{q(q+2)}}$$

For this particular choice of $\alpha$, we can compute

$$\begin{aligned}
\sum_i z_i^q/\alpha_i^q &= \left(\frac{q}{2}\right)^{-\frac{q}{q+2}} \sum_i z_i^q z_i^{-q+2/3} b_i^{2/3} \left[\sum_i z_i^{2/3} b_i^{2/3}\right]^{\frac{2q-2}{q+2}} \\
&= \left(\frac{q}{2}\right)^{-\frac{q}{q+2}} \left(\sum_i z_i^{2/3} b_i^{2/3}\right) \left[\sum_i z_i^{2/3} b_i^{2/3}\right]^{\frac{2q-2}{q+2}} \\
&= \left(\frac{q}{2}\right)^{-\frac{q}{q+2}} \left(\sum_i z_i^{2/3} b_i^{2/3}\right)^{\frac{3q}{q+2}}
\end{aligned}$$

Likewise, we can also compute

$$\begin{aligned}
\left(\sum_i \alpha_i^{2q/(3q-2)} b_i^{2q/(3q-2)}\right)^{(3q-2)/q} &= \left(\frac{q}{2}\right)^{\frac{2}{q+2}} \left(\sum_i (z_i b_i)^{2/3}\right)^{(3q-2)/q} \left(\sum_i (z_i b_i)^{2/3}\right)^{\frac{4-4q}{q(q+2)}} \\
&= \left(\frac{q}{2}\right)^{\frac{2}{q+2}} \left(\sum_i (z_i b_i)^{2/3}\right)^{\frac{3q^2+4q-4+4-4q}{q(q+2)}} \\
&= \left(\frac{q}{2}\right)^{\frac{2}{q+2}} \left(\sum_i (z_i b_i)^{2/3}\right)^{\frac{3q}{q+2}}
\end{aligned}$$

Finally, we note that $\left(\frac{q}{2}\right)^{2/(q+2)} + \left(\frac{q}{2}\right)^{-\frac{q}{q+2}} \le 2$, so we obtain

$$E^\star \le E(\alpha) \le 2\left(\sum_i (z_i b_i)^{2/3}\right)^{3q/(q+2)}$$

$\square$

# D    LOWER BOUNDING $m_F$ FOR SMOOTH LAYERS

In this section, we prove Lemma C.1, which states that when the function $F$ is a composition of functions with Lipschitz derivative, we will be able to lower bound $m_F(x,y)$ in terms of the intermediate Jacobians and layer norms evaluated at $x$. To prove Lemma C.1, we rely on tools developed by (Wei and Ma, 2019) which control the change in the output of a composition of functions if all the intermediate Jacobians are bounded.

First, we define the soft indicator $\mathbb{1}_{\le t}$ as follows:

$$\mathbb{1}_{\le t}(z) = \begin{cases} 1 & \text{if } z \le t \\ 2 - z/t & \text{if } t \le z \le 2t \\ 0 & \text{if } 2t \le z \end{cases}$$

We also define the ramp loss $T_\rho$ as follows:

$$T_\rho(z) = \begin{cases} 1 & \text{if } z \ge \rho \\ z/\rho & \text{if } 0 \le z < \rho \\ 0 & \text{if } z < 0 \end{cases}$$

Using the techniques of (Wei and Ma, 2019), we work with an "augmented" indicator which lower bounds the indicator that the prediction is correct, $\mathbb{1}[\gamma(F(x,\delta), y) \ge 0]$. We define this augmented

indicator by

$$\mathcal{I}(\delta; x, y) \triangleq T_\rho(\gamma(f_{k\leftarrow 1}(x, \delta), y)) \prod_{1 \leq i \leq k-1} \mathbb{1}_{\leq t_i}(\|f_{i\leftarrow 1}(x, \delta)\|) \prod_{1 \leq i \leq j \leq k} \mathbb{1}_{\leq \tau_{j\leftarrow i}}(\|J_{j\leftarrow i}(x, \delta)\|_{\mathrm{op}})$$

(D.1)

for nonnegative parameters $\rho, t_i, \tau_{j\leftarrow i}$ which we will later choose to be the margin, hidden layer norm, and Jacobian norms at the unperturbed input. Because the augmented indicator $\mathcal{I}(\delta; x, y)$ conditions on small Jacobian and hidden layer norms, it will turn out to be $\kappa_i^\star(x, y)$-Lipschitz in the perturbation $\delta_i$. Furthermore, by construction, the value of the augmented indicator $\mathcal{I}(\delta; x, y)$ will equal 1 when $\delta = 0$, and we will also have

$$\mathbb{1}[\gamma(F(x, \delta), y) \geq 0] \geq \mathcal{I}(\delta; x, y) \geq 1 - \sum_i \kappa_i^\star(x, y)\|\delta_i\|$$

This immediately gives a lower bound on the perturbation level required to create a negative margin. The lemma below formally bounds the Lipschitz constant of $\mathcal{I}(\delta; x, y)$ in $\delta_i$.

**Lemma D.1.** *For nonnegative parameters $t_i, \tau_{j\leftarrow i}, \rho$, with $\tau_{j\leftarrow j+1} = 1$ for any $j$ and $\tau_{j\leftarrow j'} = 0$ for $j \leq j' + 2$, define the function $\mathcal{I}(\delta; x, y)$ as in (D.1). Then in the setting of Lemma C.1, for a given $i \in [k]$, for all choices of $\delta_i$ and $\nu$, if $\delta_j = 0$ for $j > i$, we have*

$$|\mathcal{I}(\delta_i + \nu, \delta_{-i}; x, y) - \mathcal{I}(\delta_i, \delta_{-i}; x, y)| \leq \tilde{\kappa}_i\|\nu\|$$

*for $\tilde{\kappa}_i$ defined as follows:*

$$\tilde{\kappa}_i \triangleq t_{i-1}\left(\frac{8\tau_{k\leftarrow i+1}}{\rho} + \sum_{j=i}^{k-1}\frac{8\tau_{j\leftarrow i+1}}{t_j} + \sum_{1 \leq j_2 \leq j_1 \leq k}\sum_{j'=\max\{i+1, j_2\}}^{j_1} 16\kappa_{j'}' \frac{\tau_{j'-1\leftarrow i+1}\tau_{j_1\leftarrow j'+1}\tau_{j'-1\leftarrow j_2}}{\tau_{j_1\leftarrow j_2}}\right)$$

$$+ 8\sum_{j_2 \leq i \leq j_1}\frac{\tau_{j_1\leftarrow i+1}\tau_{i-1\leftarrow j_2}}{\tau_{j_1\leftarrow j_2}}$$

(D.2)

We prove Lemma D.1 in Section D.1. With Lemma D.1, we can formalize the proof of Lemma C.1.

*Proof of Lemma C.1.* We will apply Lemma D.1, using $t_i = s_i(x)$, $\rho = \gamma(F(x), y)$, $\tau_{j\leftarrow i} = \kappa_{j\leftarrow i}(x)$. First, note that for this choice of parameters, the Lipschitz constant $\tilde{\kappa}_i$ of Lemma D.1 evaluates to $\kappa_i^\star(x, y)$. Thus, it follows that for all $\delta$,

$$|\mathcal{I}(0; x, y) - \mathcal{I}(\delta; x, y)|$$
$$\leq \sum_i |\mathcal{I}(\delta_1, \ldots, \delta_{i-1}, \delta_i = 0, \delta_{j>i} = 0; x, y) - \mathcal{I}(\delta_1, \ldots, \delta_i, \delta_{j>i} = 0; x, y)|$$
$$\leq \sum_i \kappa_i^\star(x, y)\|\delta_i\|$$

(D.3)

Furthermore, by the definition of $\mathcal{I}(\delta; x, y)$, we have

$$\mathbb{1}[\gamma(F(x, \delta), y) \geq 0] \geq \mathcal{I}(\delta; x, y)$$

Finally, by our choice of the parameters used to define $\mathcal{I}(\delta; x, y)$, we also have $\mathcal{I}(0; x, y) \geq 1$. Combining everything with (D.3), we get

$$\mathbb{1}[\gamma(F(x, \delta), y) \geq 0] \geq \mathcal{I}(\delta; x, y) \geq \mathcal{I}(0; x, y) - \sum_i \kappa_i^\star(x, y)\|\delta_i\|$$
$$\geq 1 - \|\{\kappa_i^\star(x, y)/\alpha_i\}_{i=1}^k\|_{p/(p-1)}\|\{\alpha_i\|\delta_i\|\}_{i=1}^k\|_p$$
$$\text{(since } \|\cdot\|_{p/(p-1)} \text{ and } \|\cdot\|_p \text{ are dual norms)}$$
$$= 1 - \|\{\kappa_i^\star(x, y)/\alpha_i\}_{i=1}^k\|_{p/(p-1)}\|\|\delta\|\|$$

Thus, for any $\delta$, if $\|\|\delta\|\| < \|\{\kappa_i^\star(x, y)/\alpha_i\}_{i=1}^k\|_{p/(p-1)}^{-1}$, then $\mathbb{1}[\gamma(F(x, \delta), y) \geq 0] > 0$, which in turn implies $\gamma(F(x, \delta), y) \geq 0$. It follows by definition of $m_F(x, y)$ that $m_F(x, y) \geq \|\{\kappa_i^\star(x, y)/\alpha_i\}_{i=1}^k\|_{p/(p-1)}^{-1}$. $\square$

### D.1 PROOF OF LEMMA D.1

To see the core idea of the proof, consider differentiating $\mathcal{I}(\delta; x, y)$ with respect to $\delta_i$ (ignoring for the moment that the soft indicators are technically not differentiable). Let the terms $A_1, \ldots, A_q$ represent the different indicators which the product $\mathcal{I}(\delta; x, y)$ is comprised of. Then by the product rule for differentiation, we would have

$$D_{\delta_i} \mathcal{I}(\delta; x, y) = \sum_j \prod_{j' \neq j} A_{j'}(\delta; x, y) D_{\delta_i} A_j(\delta; x, y)$$

Now the idea is that for every $j$, the product $\prod_{j' \neq j} A_{j'}(\delta; x, y)$ contains an indicator that $D_{\delta_i} A_j(\delta; x, y)$ is bounded – this is stated formally by Lemmas D.3, D.4, and D.5. Informally, this allows us to bound $\|D_{\delta_i} \mathcal{I}(\delta; x, y)\|$ by the desired Lipschitz constant $\tilde{\kappa}_i$.

To formally prove this statement for the case of non-differentiable functions (as the soft-indicators $\mathbb{1}_{\leq t}$ are non-differentiable), it will be convenient to introduce the following notion of product-Lipschitzness: for functions $A_1 : \mathcal{D}_I \to \mathbb{R}^+$ and $A_2 : \mathcal{D}_I \to \mathbb{R}^+$, where $\mathcal{D}_I$ is some normed space, we say that function $A_1$ is $\bar{\tau}$-product-Lipschitz w.r.t. $A_2$ if there exists some $c, C > 0$ such that for any $\|\nu\| \leq c$ and $x \in \mathcal{D}_I$, we have

$$|A_1(x + \nu) - A_1(\nu)| A_2(x) \leq \bar{\tau} \|\nu\| + C \|\nu\|^2$$

We use the following fact that the product of functions which are product-Lipschitz with respect to one another is in fact Lipschitz. We provide the proof in Section D.2.

**Lemma D.2.** *Let $A_1, \ldots, A_q : \mathcal{D}_I \to [0, 1]$ be a set of Lipschitz functions such that $A_i$ is $\bar{\tau}_i$-product-Lipschitz w.r.t $\prod_{j \neq i} A_j$ for all $i$. Then the product $\prod_i A_i$ is $2 \sum_i \bar{\tau}_i$-Lipschitz.*

Now we proceed to formalize the intuition of product-rule differentiation presented above, by showing that the individual terms in $\mathcal{I}(\delta; x, y)$ are product-Lipschitz with respect to the other terms. For the following three lemmas, we require the technical assumption that for any fixed choice of $x, \delta_{-i}$, the functions $f_{j \leftarrow 1}(x, \delta)$, $J_{j' \leftarrow j''}(x, \delta)$ are worst-case Lipschitz in $\delta_i$ as measured in $\| \cdot \|$, $\| \cdot \|_{\mathrm{op}}$, respectively, with Lipschitz constant $C'$. Our proof of Lemma D.1, however, can easily circumvent this assumption. The proofs of the following three lemmas are given in Section D.2.

**Lemma D.3.** *Choose $i, j$ with $k - 1 \geq j \geq i$. Then after we fix any choice of $x, \delta_{-i}$, the function $\mathbb{1}_{\leq t_j}(\|f_{j \leftarrow 1}(x, \delta)\|)$ is $\frac{4\tau_{j \leftarrow i+1} t_{i-1}}{t_j}$-product-Lipschitz in $\delta_i$ with respect to $\mathbb{1}_{\leq \tau_{j \leftarrow i+1}}(\|J_{j \leftarrow i+1}(x, \delta)\|_{\mathrm{op}}) \mathbb{1}_{\leq t_{i-1}}(\|f_{i-1 \leftarrow 1}(x, \delta)\|)$.*

**Lemma D.4.** *Choose $i \leq k$. Then after we fix any choice of $x, \delta_{-i}$, the function $T_\rho(\gamma(f_{k \leftarrow 1}(x, \delta), y))$ is $\frac{4\tau_{k \leftarrow i+1} t_{i-1}}{\rho}$-product-Lipschitz in $\delta_i$ with respect to $\mathbb{1}_{\leq \tau_{k \leftarrow i+1}}(\|J_{k \leftarrow i+1}(x, \delta)\|_{\mathrm{op}}) \mathbb{1}_{\leq t_{i-1}}(\|f_{i-1 \leftarrow 1}(x, \delta)\|)$.*

**Lemma D.5.** *Choose $i, j_1, j_2$ with $j_1 \geq j_2$, $j_1 > i$. Set product-Lipschitz constant $\bar{\tau}$ as follows:*

$$\bar{\tau} = \frac{8 \left( \tau_{j_1 \leftarrow i+1} \tau_{i-1 \leftarrow j_2} + t_{i-1} \sum_{j' : \max\{j_2, i+1\} \leq j' \leq j_1} \kappa'_{j'} \tau_{j'-1 \leftarrow i+1} \tau_{j_1 \leftarrow j'+1} \tau_{j'-1 \leftarrow j_2} \right)}{\tau_{j_1 \leftarrow j_2}}$$

*Then for any fixed choice of $x, \delta_{-i}$ satisfying $\delta_j = 0$ for $j > i$, the function $\mathbb{1}_{\leq \tau_{j_1 \leftarrow j_2}}(\|J_{j_1 \leftarrow j_2}(x, \delta)\|_{\mathrm{op}})$ is $\bar{\tau}$-product-Lipschitz in $\delta_i$ with respect to $\mathbb{1}_{\leq t_{i-1}}(\|f_{i-1 \leftarrow 1}(x, \delta)\|) \prod_{1 \leq j'' \leq j' \leq k} \mathbb{1}_{\leq \tau_{j' \leftarrow j''}}(\|J_{j' \leftarrow j''}(x, \delta)\|_{\mathrm{op}})$. Here note that we have $\tau_{i-1 \leftarrow j_2} = 0$ if $i - 1 \leq j_2 + 1$.*

Given the described steps, we will now complete the proof of Lemma D.1.

*Proof of Lemma D.1.* We first assume that the conditions of Lemmas D.3, D.4, D.5 regarding $C'$-worst-case Lipschitzness hold. We note that $\mathcal{I}(\delta; x, y)$ is a product which contains all the functions appearing in Lemmas D.3, D.4, and D.5. Thus, Claim D.1 allows us to conclude that each term in $\mathcal{I}(\delta; x, y)$ is product-Lipschitz with respect to the product of the remaining terms. As these lemmas also account for all the terms in $\mathcal{I}(\delta; x, y)$, we can thus apply Lemma D.2, where each $A_i$ is set to be a term in the product for $\mathcal{I}(\delta; x, y)$. Therefore, to bound the Lipschitz constant in $\delta_i$ of $\mathcal{I}(\delta; x, y)$, we sum the product-Lipschitz constants given by Lemmas D.3, D.4, and D.5. This gives that $\mathcal{I}(\delta; x, y)$ is $\tilde{\kappa}_i$-Lipschitz in $\delta_i$ for $\tilde{\kappa}_i$ defined in (D.2).

Now to remove the $C'$ worst-case Lipschitzness assumption, we can follow the reasoning of Claim D.6 of (Wei and Ma, 2019) to note that such Lipschitz constants exist if we restrict $\delta_i$ to some compact set, and thus conclude the lemma statement for $\delta_i$ restricted to this compact set. Now we simply choose this compact set sufficiently large to include both $\delta_i$ and $\delta_i + \nu$. $\qquad\square$

## D.2 PROOFS FOR PRODUCT-LIPSCHITZ LEMMAS

*Proof of Lemma D.2.* As each $A_i$ is Lipschitz and there are a finite number of functions, there exists $C'$ such that any possible product $A_{i_1} A_{i_2} \cdots A_{i_j}$ is $C'$-Lipschitz. Furthermore, by the definition of product-Lipschitz, there are $c, C > 0$ such that for any $\|\nu\| \leq c$, $x \in \mathcal{D}_I$, and $1 \leq i \leq q$, we have

$$|(A_i(x+\nu) - A_i(x))| \prod_{j \neq i} A_j(x) \leq \bar{\tau}_i \|\nu\| + C\|\nu\|^2$$

Now we note that

$$\prod_i A_i(x+\nu) - \prod_i A_i(x) = \sum_i \left( \prod_{j=1}^{i-1} A_j(x+\nu) \prod_{j=i+1}^{q} A_j(x) \right) (A_i(x+\nu) - A_i(x)) \quad \text{(D.4)}$$

Now for any $i$, we have

$$|(A_i(x+\nu) - A_i(x)) \prod_{j=1}^{i-1} A_j(x+\nu) \prod_{j=i+1}^{q} A_j(x)|$$

$$\leq |(A_i(x+\nu) - A_i(x))|(\prod_{j=1}^{i-1} A_j(x) + C'\|\nu\|) \prod_{j=i+1}^{q} A_j(x) \quad \text{(as } \prod_{j=1}^{i-1} A_j \text{ is } C'\text{-Lipschitz)}$$

$$\leq |A_i(x+\nu) - A_i(x)|(\prod_{j \neq i} A_j(x) + C'\|\nu\|)$$

We used the fact that $\prod_{j=i+1}^{q} A_j(x) \leq 1$. Now we have $|A_i(x+\nu) - A_i(x)| \prod_{j \neq i} A_j(x) \leq \bar{\tau}_i \|\nu\| + C\|\nu\|^2$, and $|A_i(x+\nu) - A_i(x)|C'\|\nu\| \leq C'^2\|\nu\|^2$ as $A_i$ is $C'$-Lipschitz, so

$$|(A_i(x+\nu) - A_i(x)) \prod_{j=1}^{i-1} A_j(x+\nu) \prod_{j=i+1}^{q} A_j(x)| \leq \bar{\tau}_i \|\nu\| + (C + C'^2)\|\nu\|^2$$

Plugging this back into (D.4) and applying triangle inequality, we get

$$|\prod_i A_i(x+\nu) - \prod_i A_i(x)| \leq \|\nu\| \left( \sum_i \bar{\tau}_i + q(C + C'^2)\|\nu\| \right)$$

Define the constant $C'' \triangleq \min\{c, \frac{\sum_i \bar{\tau}_i}{q(C+C'^2)}\}$. For any $x$ and all $\nu$ satisfying $\|\nu\| \leq C''$, we have

$$|\prod_i A_i(x+\nu) - \prod_i A_i(x)| \leq 2\|\nu\| \sum_i \bar{\tau}_i \quad \text{(D.5)}$$

Now for any $x, y \in \mathcal{D}_I$, we wish to show $|\prod_i A_i(y) - \prod_i A_i(x)| \leq 2\|x - y\| \sum_i \bar{\tau}_i$. To this end, we divide $x - y$ into segments of length at most $C''$ and apply (D.5) on each segment.

Define $x^{(j)} = x + j\frac{C''}{\|x-y\|}(y-x)$ for $j = 1, \ldots, \lfloor \|x-y\|/C'' \rfloor$. Then as $\|x^{(j)} - x^{(j-1)}\| \leq C''$, we have $|\prod_i A_i(x^{(j)}) - \prod_i A_i(x^{(j-1)})| \leq \|x^{(j)} - x^{(j-1)}\| \sum_i \bar{\tau}_i$. Furthermore, we note that the sum of all the segment lengths equals $\|y - x\|$. Thus, we can sum this inequality over pairs $(x, x^{(1)}), \ldots, (x^{(\lfloor \|x-y\|/C'' \rfloor)}, y)$ and apply triangle inequality to get

$$|\prod_i A_i(y) - \prod_i A_i(x)| \leq 2\|x - y\| \sum_i \bar{\tau}_i$$

as desired. $\qquad\square$

*Proof of Lemma D.3.* For convenience, define

$$A(x, \delta) \triangleq \mathbb{1}_{\leq \tau_{j \leftarrow i+1}}(\|J_{j \leftarrow i+1}(x, \delta)\|_{\mathrm{op}}) \mathbb{1}_{\leq t_{i-1}}(\|f_{i-1 \leftarrow 1}(x, \delta)\|)$$

We first note that $D_{\delta_i} f_{j \leftarrow 1}(x, \delta)$, the partial derivative of $f_{j \leftarrow 1}$ with respect to $\delta_i$, is given by $J_{j \leftarrow i+1}(x, \delta) \|f_{i-1 \leftarrow 1}(x, \delta)\|$ by Claim D.2. As $J_{j \leftarrow i+1}(x, \delta)$, $\|f_{i-1 \leftarrow 1}(x, \delta)\|$ are both worst-case Lipschitz in $\delta_i$ with some Lipschitz constant $C'$, we can apply Claim H.4 of (Wei and Ma, 2019) to obtain:

$$\|f_{j \leftarrow 1}(x, \delta_{-i}, \delta_i + \nu) - f_{j \leftarrow 1}(x, \delta_{-i}, \delta_i)\| \leq (\|D_{\delta_i} f_{j \leftarrow 1}(x, \delta)\|_{\mathrm{op}} + C''/2\|\nu\|)\|\nu\|$$

for any $\nu$ and some Lipschitz constant $C''$. Thus, by the $t_j^{-1}$ Lipschitz-ness of the indicator $\mathbb{1}_{\leq t_j}$, we have

$$|\mathbb{1}_{\leq t_j}(\|f_{j \leftarrow 1}(x, \delta_{-i}, \delta_i + \nu)\|) - \mathbb{1}_{\leq t_j}(\|f_{j \leftarrow 1}(x, \delta_{-i}, \delta_i\|)|A(x, \delta)$$
$$\leq A(x, \delta)\frac{\|f_{j \leftarrow 1}(x, \delta_{-i}, \delta_i + \nu) - f_{j \leftarrow 1}(x, \delta_{-i}, \delta_i)\|}{t_j}$$
$$\leq A(x, \delta)\frac{(\|D_{\delta_i} f_{j \leftarrow 1}(x, \delta)\|_{\mathrm{op}} + C''/2\|\nu\|)\|\nu\|}{t_j}$$
$$\leq A(x, \delta)\frac{(\|J_{j \leftarrow i+1}(x, \delta)\|_{\mathrm{op}}\|f_{i-1 \leftarrow 1}(x, \delta)\| + C''/2\|\nu\|)\|\nu\|}{t_j}$$

Now by definition of $A(x, \delta)$, we get that the right hand side equals 0 if $\|J_{j \leftarrow i+1}(x, \delta)\|_{\mathrm{op}} \geq 2\tau_{j \leftarrow i+1}$ or $\|f_{i-1 \leftarrow 1}(x, \delta)\| \geq 2t_{i-1}$. Thus, the right hand side must be bounded by

$$\frac{4\tau_{j \leftarrow i+1} t_{i-1}}{t_j}\|\nu\| + C''/2\|\nu\|^2$$

which gives product-Lipschitzness with constant $\frac{4\tau_{j \leftarrow i+1} t_{i-1}}{t_j}$. $\qquad\square$

*Proof of Lemma D.4.* This proof follows in an identical manner to that of Lemma D.3. The only additional step is using the fact that $\gamma(h, y)$ is 1-Lipschitz in $h$, so the composition $T_\rho(\gamma(h, y))$ is $\rho^{-1}$-Lipschitz in $h$. $\qquad\square$

*Proof of Lemma D.5.* Let $C'$ be an upper bound on the Lipschitz constant in $\delta_i$ of all the functions $J_{j' \leftarrow j''}(x, \delta)$. As we assumed that each $f_j$ has $\kappa'_j$-Lipschitz Jacobian, such an upper bound exists. We first argue that

$$\|J_{j_1 \leftarrow j_2}(x, \delta_{-i}, \delta_i + \nu) - J_{j_1 \leftarrow j_2}(x, \delta_{-i}, \delta_i)\|_{\mathrm{op}}$$
$$\leq \|\nu\| \sum_{j' : \max\{j_2, i+1\} \leq j' \leq j_1} \Big( \kappa'_{j'}(\|J_{j_1 \leftarrow j'+1}(x, \delta)\|_{\mathrm{op}} + C'/2\|\nu\|)$$
$$\cdot (\|J_{j'-1 \leftarrow i+1}(x, \delta)\|_{\mathrm{op}}\|f_{i-1 \leftarrow 1}(x, \delta)\| + C''/2\|\nu\|)\|J_{j'-1 \leftarrow j_2}(x, \delta)\|_{\mathrm{op}} \Big)$$
$$+ \|\nu\|(\|J_{j_1 \leftarrow i+1}\|_{\mathrm{op}} + C'/2\|\nu\|)\|J_{i-1 \leftarrow j_2}\|_{\mathrm{op}}$$

(D.6)

for some Lipschitz constant $C''$. The proof of this statement is nearly identical to the proof of Claim D.3 in (Wei and Ma, 2019), so we only sketch it here and point out the differences. We rely on the expansion

$$J_{j_1 \leftarrow j_2}(x, \delta) = J_{j_1 \leftarrow j_1}(x, \delta) J_{j_1 - 1 \leftarrow j_1 - 1}(x, \delta) \cdots J_{j_2 \leftarrow j_2}(x, \delta)$$

which follows from the chain rule. Now we note that we can compute the change in a single term $J_{j' \leftarrow j'}(x, \delta)$ from perturbing $\delta_i$ as follows:

$$\|J_{j' \leftarrow j'}(x, \delta_{-i}, \delta_i + \nu) - J_{j' \leftarrow j'}(x, \delta_{-i}, \delta_i)\|_{\mathrm{op}}$$
$$= \|D_h f_{j' \leftarrow j'}(h, \delta)|_{h = f_{j'-1 \leftarrow 1}(x, \delta_{-i}, \delta_i + \nu)} - D_h f_{j' \leftarrow j'}(h, \delta)|_{h = f_{j'-1 \leftarrow 1}(x, \delta_{-i}, \delta_i)}\|_{\mathrm{op}}$$

Note that when $j' > i$, by assumption $\delta_{j'} = 0$, so $D_h f_{j' \leftarrow j'}(h, \delta) = D_h f_{j'}(h)$. Thus, as $f_{j'}$ has $\kappa'_{j'}$-Lipschitz derivative, we get

$$\|J_{j' \leftarrow j'}(x, \delta_{-i}, \delta_i + \nu) - J_{j' \leftarrow j'}(x, \delta_{-i}, \delta_i)\|_{\text{op}}$$
$$\leq \kappa'_{j'} \|f_{j'-1 \leftarrow 1}(x, \delta_{-i}, \delta_i + \nu) - f_{j'-1 \leftarrow 1}(x, \delta_{-i}, \delta_i)\|$$
$$\qquad\qquad\qquad\qquad\qquad \text{(since the derivative of } f_{j'} \text{ is } \kappa'_{j'}\text{-Lipschitz)}$$
$$\leq \kappa'_{j'} (\|D_{\delta_i} f_{j'-1 \leftarrow 1}(x, \delta_{-i}, \delta_i)\|_{\text{op}} + C''/2\|\nu\|)\|\nu\| \quad \text{(by Claim H.4 of (Wei and Ma, 2019))}$$
$$\leq \kappa'_{j'} (\|J_{j'-1 \leftarrow i+1}(x, \delta)\|_{\text{op}} \|f_{i-1 \leftarrow 1}(x, \delta)\| + C''/2\|\nu\|)\|\nu\|$$

We obtained the last line via Claim D.2. We note that the cases when $j' > i$ contribute to the terms under the summation in (D.6).

When $j' = i$, we have $D_h f_{i \leftarrow i}(h, \delta) = D_h f_i(h) + D_h[\delta_i\|h\|]$ for any $h$, so $\|D_h f_{i \leftarrow i}(h, \delta_i, \delta_{-i}) - D_h f_{i \leftarrow i}(h, \delta_i + \nu, \delta_{-i})\|_{\text{op}} = \|D_h[\nu\|h\|]\|_{\text{op}} \leq \|\nu\|$. As this holds for any $h$, it follows that $\|J_{i \leftarrow i}(x, \delta_{-i}, \delta_i + \nu) - J_{i \leftarrow i}(x, \delta_{-i}, \delta_i)\|_{\text{op}} \leq \|\nu\|$. This term results in the last quantity in (D.6). Finally, when $j' < i$, we have $J_{j' \leftarrow j'}(x, \delta_{-i}, \delta_i + \nu) = J_{j' \leftarrow j'}(x, \delta_{-i}, \delta_i)$ as $J_{j' \leftarrow j'}$ does not depend on $\delta_i$.

To see how (D.6) follows, we would apply the above bounds in a telescoping sum over indices $j'$ ranging from $\max\{j_2, i\}$ to $j_1$. For a more detailed derivation, refer to the steps in Claim D.3 of (Wei and Ma, 2019).

Now for convenience define

$$A(x, \delta) \triangleq \mathbb{1}_{\leq t_{i-1}}(\|f_{i-1 \leftarrow 1}(x, \delta)\|) \prod_{1 \leq j'' \leq j' \leq k} \mathbb{1}_{\leq \tau_{j' \leftarrow j''}}(\|J_{j' \leftarrow j''}(x, \delta)\|_{\text{op}})$$

Note that if any of the bounds set by the indicators in $A(x, \delta)$ are violated, then $A(x, \delta) = 0$, and thus

$$|\mathbb{1}_{\leq \tau_{j_1 \leftarrow j_2}}(\|J_{j_1 \leftarrow j_2}(x, \delta_{-i}, \delta_i + \nu)\|_{\text{op}}) - \mathbb{1}_{\leq \tau_{j_1 \leftarrow j_2}}(\|J_{j_1 \leftarrow j_2}(x, \delta_{-i}, \delta_i)\|_{\text{op}})|A(x, \delta) = 0$$

In the other case, we have $\|f_{i-1 \leftarrow 1}(x, \delta)\| \leq 2t_{i-1}$, and $\|J_{j' \leftarrow j''}(x, \delta)\|_{\text{op}} \leq 2\tau_{j' \leftarrow j''}$, in which case (D.6) can be bounded by

$$\|J_{j_1 \leftarrow j_2}(x, \delta_{-i}, \delta_i + \nu) - J_{j_1 \leftarrow j_2}(x, \delta_{-i}, \delta_i)\|_{\text{op}} A(x, \delta)$$
$$\leq 8t_{i-1} \left( \sum_{j' : \max\{j_2, i+1\} \leq j' \leq j_1} \kappa'_{j'} \tau_{j'-1 \leftarrow i+1} \tau_{j_1 \leftarrow j'+1} \tau_{j'-1 \leftarrow j_2} \right) \|\nu\|$$
$$+ \tau_{j_1 \leftarrow i+1} \tau_{i-1 \leftarrow j_2} \|\nu\| + C'''\|\nu\|^2$$

for some $C'''$ that is independent of $x, \delta, \nu$. Thus, by Lipschitz-ness of $\mathbb{1}_{\leq \tau_{j_1 \leftarrow j_2}}(\cdot)$ and the triangle inequality, we have

$$|\mathbb{1}_{\leq \tau_{j_1 \leftarrow j_2}}(\|J_{j_1 \leftarrow j_2}(x, \delta_{-i}, \delta_i + \nu)\|_{\text{op}}) - \mathbb{1}_{\leq \tau_{j_1 \leftarrow j_2}}(\|J_{j_1 \leftarrow j_2}(x, \delta_{-i}, \delta_i)\|_{\text{op}})|A(x, \delta) \leq$$
$$\tag{D.7}$$

$$\frac{8\left(\tau_{j_1 \leftarrow i+1} \tau_{i-1 \leftarrow j_2} + t_{i-1} \sum_{j' : \max\{j_2, i+1\} \leq j' \leq j_1} \kappa'_{j'} \tau_{j'-1 \leftarrow i+1} \tau_{j_1 \leftarrow j'+1} \tau_{j'-1 \leftarrow j_2}\right) \|\nu\|}{\tau_{j_1 \leftarrow j_2}} + C'''\|\nu\|^2$$

This gives the desired result. $\qquad\qquad\qquad\qquad\qquad\qquad\qquad\qquad\qquad\qquad\qquad\quad\square$

**Claim D.1.** *Let $A_1, A_2, A_3 : \mathcal{D}_I \to [0, 1]$ be functions where $A_1$ is $\bar{\tau}$-product-Lipschitz w.r.t. $A_2$. Then $A_1$ is also $\bar{\tau}$-product Lipschitz w.r.t. $A_2 A_3$.*

*Proof.* This statement follows from the definition of product-Lipschitzness and the fact that

$$|A_1(x + \nu) - A_1(\nu)|A_2(x)A_3(x) \leq |A_1(x + \nu) - A_1(\nu)|A_2(x)$$

since $A_3(x) \leq 1$. $\qquad\qquad\qquad\qquad\qquad\qquad\qquad\qquad\qquad\qquad\qquad\qquad\qquad\qquad\quad\square$

**Claim D.2.** *The partial derivative of $f_{j \leftarrow 1}$ with respect to variable $\delta_i$ evaluated at $x, \delta$ can be computed as*

$$D_{\delta_i} f_{j \leftarrow 1}(x, \delta) = J_{j \leftarrow i+1}(x, \delta)\|f_{i-1 \leftarrow 1}(x, \delta)\|$$

*Proof.* By definition, $f_{j\leftarrow 1}(x,\delta) = f_{j\leftarrow i+1}(f_{i\leftarrow 1}(x,\delta),\delta)$. Thus, we note that $f_{j\leftarrow 1}(x,\delta)$ only depends on $\delta_i$ through $f_{i\leftarrow 1}(x,\delta)$, so by chain rule we have

$$
\begin{aligned}
D_{\delta_i} f_{j\leftarrow 1}(x,\delta) &= D_h f_{j\leftarrow i+1}(h,\delta)|_{h=f_{i\leftarrow 1}(x,\delta)} D_{\delta_i} f_{i\leftarrow 1}(x,\delta) \\
&= J_{j\leftarrow i+1}(x,\delta) D_{\delta_i}[f_i(f_{i-1\leftarrow 1}(x,\delta)) + \delta_i \| f_{i-1\leftarrow 1}(x,\delta)\|] \\
&= J_{j\leftarrow i+1}(x,\delta)\| f_{i-1\leftarrow 1}(x,\delta)\|
\end{aligned}
$$

In the second line, we invoked the definition of $J_{j\leftarrow i+1}(x,\delta)$. $\square$

**Claim D.3.** *We have the expansion*

$$
J_{j\leftarrow i}(x,\delta) = J_{j\leftarrow j}(x,\delta)\cdots J_{i\leftarrow i}(x,\delta)
$$

*Proof.* This is a result of the chain rule, but for completeness we state the proof here. We have

$$
\begin{aligned}
D_h f_{j\leftarrow i}(h,\delta) &= D_h f_j(f_{j-1\leftarrow i}(h,\delta),\delta) \\
&= D_{h'} f_{j\leftarrow j}(h',\delta)|_{h'=f_{j-1\leftarrow i}(h,\delta)} D_h f_{j-1\leftarrow i}(h,\delta) \qquad \text{(by chain rule)}
\end{aligned}
$$

$$(D.8)$$

Thus, plugging in $h = f_{i-1\leftarrow 1}(x,\delta)$, we get

$$
\begin{aligned}
J_{j\leftarrow i}(x,\delta) &= D_h f_{j\leftarrow i}(h,\delta)|_{h=f_{i-1\leftarrow 1}(x,\delta)} \\
&= D_{h'} f_{j\leftarrow j}(h',\delta)|_{h'=f_{j-1\leftarrow i}(f_{i-1\leftarrow 1}(x,\delta),\delta)} D_h f_{j-1\leftarrow i}(h,\delta)|_{h=f_{i-1\leftarrow 1}(x,\delta)} \\
&= D_{h'} f_{j\leftarrow j}(h',\delta)|_{h'=f_{j-1\leftarrow 1}(x,\delta)} J_{j-1\leftarrow i}(x,\delta) \\
&= J_{j\leftarrow j}(x,\delta) J_{j-1\leftarrow i}(x,\delta)
\end{aligned}
$$

Now we can apply identical steps to expand $J_{j-1\leftarrow i}(x,\delta)$, giving the desired result. $\square$

## E    PROOFS FOR ADVERSARIALLY ROBUST CLASSIFICATION

In this section, we derive the generalization bounds for adversarial classification presented in Section 4. Recall the adversarial all-layer margin $m_F^{\text{adv}}(x,y) \triangleq \min_{x'\in\mathcal{B}^{\text{adv}}(x)} m_F(x',y)$ defined in Section 4. In this section, we will use the general definition of $m_F$ in (A.1).

We will sketch the proof of Theorem E.1 using the same steps as those laid out in Sections 2 and A. We will rely on the following general analogue of Theorem C.1 with the exact same proof, but with $\kappa_i^\star$ (defined in Section C) replaced by $\kappa_i^{\text{adv}}(x,y) \triangleq \max_{x'\in\mathcal{B}^{\text{adv}}(x)} \kappa_i^\star(x',y)$ everywhere:

**Theorem E.1.** *Let $\mathcal{F} = \{f_k \circ \cdots \circ f_1 : f_i \in \mathcal{F}_i\}$ denote a class of compositions of functions from $k$ families $\{\mathcal{F}_i\}_{i=1}^k$, each of which satisfies Condition A.1 with operator norm $\|\cdot\|_{\text{op}}$ and complexity $\mathcal{C}_{\|\cdot\|_{\text{op}}}(\mathcal{F}_i)$. For any choice of integer $q > 0$, with probability $1-\delta$ for all $F \in \mathcal{F}$ the following bound holds:*

$$
\mathbb{E}_P[\ell_{0\text{-}1}^{\text{adv}}(F(x),y)]
$$

$$
\leq \frac{3}{2}\left(\mathbb{E}_{P_n}[\ell_{0\text{-}1}^{\text{adv}}(F(x),y)]\right) + O\left(\frac{k\log n + \log(1/\delta)}{n}\right)
$$

$$
+ \left(1 - \mathbb{E}_{P_n}[\ell_{0\text{-}1}^{\text{adv}}(F(x),y)]\right)^{\frac{2}{q+2}} O\left(q\left(\frac{\log^2 n}{n}\right)^{\frac{q}{q+2}}\left(\sum_i \|\kappa_i^{\text{adv}}\|_{L_q(\mathcal{S}_n^{\text{adv}})}^{2/3} \mathcal{C}_{\|\cdot\|_{\text{op}}}(\mathcal{F}_i)^{2/3}\right)^{\frac{3q}{q+2}}\right)
$$

*where $\mathcal{S}_n^{\text{adv}}$ denotes the subset of training examples correctly classified by $F$ with respect to adversarial perturbations. In particular, if $F$ classifies all training samples with adversarial error 0, i.e. $|\mathcal{S}_n^{\text{adv}}| = n$, with probability $1-\delta$ we have*

$$
\mathbb{E}_P[\ell_{0\text{-}1}^{\text{adv}}(F(x),y)] \lesssim q\left(\frac{\log^2 n}{n}\right)^{q/(q+2)}\left(\sum_i \|\kappa_i^{\text{adv}}\|_{L_q(\mathcal{S}_n^{\text{adv}})}^{2/3} \mathcal{C}_{\|\cdot\|_{\text{op}}}(\mathcal{F}_i)^{2/3}\right)^{3q/(q+2)} + \zeta
$$

*where $\zeta \triangleq O\left(\frac{k\log n + \log(1/\delta)}{n}\right)$ is a low order term.*

Given Theorem E.1, Theorem 4.1 follows with the same proof as the proof of Theorem 3.1 given in Section B. To prove Theorem E.1, we first have the following analogue of Lemma A.1 bounding the covering number of $m^{\mathrm{adv}} \circ \mathcal{F}$.

**Lemma E.1.** *Define $m^{\mathrm{adv}} \circ \mathcal{F} \triangleq \{(x, y) \mapsto m_F^{\mathrm{adv}}(x, y) : F \in \mathcal{F}\}$. Then*

$$\mathcal{N}_\infty(\epsilon, m^{\mathrm{adv}} \circ \mathcal{F}) \le \mathcal{N}_{\|\!|\cdot|\!\|}(\epsilon, \mathcal{F})$$

This lemma allows us to invoke Lemma A.2 on a smooth loss composed with $m^{\mathrm{adv}} \circ \mathcal{F}$, as we did for the clean classification setting. The lemma is proven the exact same way as Lemma A.1, given the Lipschitz-ness of $m_F^{\mathrm{adv}}$ below:

**Claim E.1.** *For any $x, y \in \mathcal{D}_0 \times [l]$, and function sequences $F = \{f_i\}_{i=1}^k, \widehat{F} = \{\widehat{f}_i\}_{i=1}^k$, we have $|m_F^{\mathrm{adv}}(x, y) - m_{\widehat{F}}^{\mathrm{adv}}(x, y)| \le \|\!|F - \widehat{F}|\!\|.$*

*Proof.* Let $x^\star \in \mathcal{B}^{\mathrm{adv}}(x)$ be such that $m_F^{\mathrm{adv}}(x, y) = m_F(x^\star, y)$. By Claim A.1, we have

$$m_{\widehat{F}}^{\mathrm{adv}}(x, y) \le m_{\widehat{F}}(x^\star, y) \le m_F(x^\star, y) + \|\!|F - \widehat{F}|\!\| = m_F^{\mathrm{adv}}(x, y) + \|\!|F - \widehat{F}|\!\|$$

We can apply the reverse reasoning to also obtain $m_F^{\mathrm{adv}}(x, y) \le m_{\widehat{F}}^{\mathrm{adv}}(x, y) + \|\!|F - \widehat{F}|\!\|$. Combining the two gives us the desired result. $\qquad\square$

Next, we lower bound $m_F^{\mathrm{adv}}$ when each function in $F$ is smooth.

**Lemma E.2.** *In the setting of Lemma C.1, let $\kappa_i^{\mathrm{adv}}(x, y) \triangleq \max_{x' \in \mathcal{B}^{\mathrm{adv}}(x)} \kappa_i^\star(x', y)$. Then if $\ell_{0\text{-}1}^{\mathrm{adv}}(F(x), y) = 0$, we have*

$$m_F^{\mathrm{adv}}(x, y) \ge \|\{\kappa_i^{\mathrm{adv}}(x, y)/\alpha_i\}_{i=1}^k\|_{p/(p-1)}^{-1}$$

*Proof.* By definition and Lemma C.1, we have

$$
\begin{aligned}
m_F^{\mathrm{adv}}(x, y) = \min_{x' \in \mathcal{B}^{\mathrm{adv}}(x)} m_F(x', y) &\ge \min_{x' \in \mathcal{B}^{\mathrm{adv}}(x)} \|\{\kappa_i^\star(x', y)/\alpha_i\}_{i=1}^k\|_{p/(p-1)}^{-1} \\
&= \frac{1}{\max_{x' \in \mathcal{B}^{\mathrm{adv}}(x)} \|\{\kappa_i^\star(x', y)/\alpha_i\}_{i=1}^k\|_{p/(p-1)}} \\
&\ge \frac{1}{\|\{\max_{x' \in \mathcal{B}^{\mathrm{adv}}(x)} \kappa_i^\star(x', y)/\alpha_i\}_{i=1}^k\|_{p/(p-1)}} \\
&= \frac{1}{\|\{\kappa_i^{\mathrm{adv}}(x', y)/\alpha_i\}_{i=1}^k\|_{p/(p-1)}}
\end{aligned}
$$

$\qquad\square$

To finish the proof of Theorem E.1, we use $\ell_{\beta=1}(m_F^{\mathrm{adv}}(x, y))$ as an upper bound for $\ell_{0\text{-}1}^{\mathrm{adv}}(F(x), y)$ and follow the steps of Theorem C.1 to optimize over the choice of $\{\alpha_i\}_{i=1}^k$. As these steps are identical to Theorem C.1, we omit them here. With Theorem E.1 in hand, we can conclude Theorem 4.1 using the same proof as Theorem 3.1.

## F  ADDITIONAL EXPERIMENTAL DETAILS

### F.1  ADDITIONAL DETAILS FOR CLEAN CLASSIFICATION SETTING

For the experiments presented in Table 1, the other hyperparameters besides $t$ and $\eta_{\mathrm{perturb}}$ are set to their defaults for WideResNet architectures. Our code is inspired by the following PyTorch WideResNet implementation: https://github.com/xternalz/WideResNet-pytorch, and we use the default hyperparameters in this implementation. Although we tried a larger choice of $t$, the number of updates to the perturbations $\delta$, our results did not depend much on $t$.

Table 3: Validation error on CIFAR-10 for VGG-19 architecture trained with standard SGD and AMO.

| Arch. | Standard SGD | AMO |
|---|---|---|
| VGG-19 | 5.66% | **5.06%** |

Table 4: Validation error on CIFAR for models trained with dropout vs. AMO. We tuned the dropout probability $p$ and display the best-performing value.

| Dataset | Arch. | Tuned Dropout | AMO |
|---|---|---|---|
| CIFAR-10 | WRN16-10 | 4.04% ($p = 0.1$) | **3.42%** |
| | WRN28-10 | 3.52% ($p = 0.1$) | **3.00%** |
| CIFAR-100 | WRN16-10 | 19.57% ($p = 0.1$) | **19.14%** |
| | WRN28-10 | 18.77% ($p = 0.2$) | **17.78%** |

In Table 3, we demonstrate that our AMO algorithm can also improve performance for conventional feedforward architectures such as VGG (Simonyan and Zisserman, 2014). We report the best validation error on the clean classification setting. For both methods, we train VGG-19 for 350 epochs using weight decay 0.0005 and an initial learning rate of 0.05 annealing by a factor of 0.1 at the 150-th and 250-th epochs. For the AMO algorithm, we optimize perturbations $\delta$ for $t = 1$ steps and use learning rate $\eta_{\text{perturb}} = 0.01$.

In Table 4, we demonstrate that our AMO algorithm can offer improvements over dropout, which is also a regularization method based on perturbing the hidden layers. This demonstrates the value of regularizing stability with respect to *worst-case* perturbations rather than random perturbations. For the combinations of CIFAR dataset and WideResNet architecture, we tuned the level of dropout and display the best tuned results with the dropout probability $p$ in Table 4. We use the same hyperparameter settings as described above.

## F.2 ADDITIONAL DETAILS FOR ROBUST AMO

Our implementation is based on the robustness library by (Engstrom et al., 2019)[5]. For both methods, we produce the perturbations during training using 10 steps of PGD with $\ell_\infty$ perturbations with radius $\epsilon = 8$. We update the adversarially perturbed training inputs with a step size of 2 in pixel space. During evaluation, we use 50 steps of PGD with 10 random restarts and perturbation radius $\epsilon = 8$. For all models, we train for 150 epochs using SGD with a learning rate decay of 0.1 at the 50th, 100th, and 150th epochs. For the baseline models, we use a weight decay of 5e-4, initial learning rate of 0.1, and batch size of 128.

For both models trained with robust AMO, the perturbations $\delta$ have a step size of 6.4e-3, and we shrink the norm of $\delta$ by a factor of 0.92 every iteration. For the WideResNet16 model trained with robust AMO, we use a batch size of 128 with initial learning rate of 0.1 and weight decay 5e-4. For the WideResNet28 model trained using robust AMO, we use a batch size of 64 (chosen so that the $\delta$ vectors fit in GPU memory) with initial learning rate of 0.707 (chosen to keep the covariance of the gradient update the same with the decreased batch size). We use weight decay 2e-4. We chose these parameters by tuning.

---

[5] https://github.com/MadryLab/robustness

