# OpenReview forum: "Improved Sample Complexities for Deep Neural Networks and Robust Classification via an All-Layer Margin"
_ICLR.cc/2020/Conference — Accept (Poster)_

### Official Review · AnonReviewer1 · 2019-10-22
**Official Blind Review #1**

**Rating:** 3

**Review:**

This paper proves generalization bounds for Neural networks in terms of all layer margins. All layer margins are the smallest relative perturbations  of outputs of each layer, that result in misclassification. This new quantity allows to show generalization bounds that scales as sum of complexities of each layer and inversely scales with the all layer margin. The resulting bound does not have an explicit exponential dependence on depth, unlike some earlier bounds in terms of output margin.

The paper makes strong claims in the abstract and introduction that the analysis removes exponential dependency of capacity on depth. However I don’t think this is possible in the worst case. The exponential dependency is now hidden in the all layer margin (please correct me if I am mistaken). Note that the bound could still be interesting if this quantity is small for real world networks of interest. However the paper currently lacks this discussion. The paper need to make it clear that the bound only avoids explicit exponential dependence on depth. This brings me to another major drawback of the all layer margin.

This quantity is not computable as it requires solving a min-max problem, to find the best perturbation for each data point. So we cannot easily test if real networks that generalize well do have smaller all layer margin. The paper provides a lower bound for this quantity in terms of layer norms, network Jacobian norms, and output margin - which can be large. While all layer margin is certainly interesting from analysis perspective, as it allows to show bounds that are relatively cleaner, and behaves well with composition functions, it is not clear to me that it doesn’t decrease exponentially for deeper networks. A discussion on this can help the paper.

This paper is along the recent line of work on data dependent generalization bounds by Nagarajan  & Kolter 2019, Wei & Ma 2019.  Note that these earlier papers also prove generalization bounds that also do not have explicit exponential dependence on depth. This point also needs to be made clearer.

Given the adversarial nature of the definition of the all layer margin, the same framework is used to show a robust generalization bound, which is nice. However, the bound again depends on a hard to compute quantity - \kapp^{adv}, making it less clear about the utility of such a bound.

Finally the paper presents experiments, where to encourage higher all layer margin, the paper relies on min max gradient updates, similar to adversarial training, and shows better performance with this. I am curious if authors experiments with dropout, as it also adds “perturbations” to the output activations of each layer, and if the gains are orthogonal to dropout. Also how does the performance of AMO compared with  adversarial training methods (PGD) with small \epsilon? The experiments currently are lacking to compare AMO with existing techniques.

Overall I believe the paper needs to be rewritten with correctly stated contributions and need a more nuanced discussion of the benefits and drawbacks of the all layer margin and the  corresponding generalization bounds.

** Post response comments **
Authors have clarified some of my earlier concerns. But the paper still requires rewriting to properly emphasize that it does not avoid exponential dependence on depth, but rather avoids an explicit dependence. Also there needs to be discussion about the computability of  \kapp^{adv}. Finally adding experiments with dropout should make it more clear about the advantages of the proposed training method. Because the current draft does not address these yet, I am leaving my score at 3. My actual rating is more on the borderline (5), but the current system does not allow such scores.


**Experience Assessment:**

I have published in this field for several years.

**Review Assessment: Checking Correctness Of Derivations And Theory:**

I assessed the sensibility of the derivations and theory.

**Review Assessment: Checking Correctness Of Experiments:**

I did not assess the experiments.

**Review Assessment: Thoroughness In Paper Reading:**

I read the paper thoroughly.

---

> ### Author Response · Authors · 2019-11-15
> **Response**
>
> We thank the reviewer for the comments, which we will incorporate into the final version of our paper.
>
> A concern raised in the review is that our generalization bounds could still suffer severe (exponential) worst-case dependencies on depth. This concern also applies to the prior work of [Wei and Ma’19, Nagarajan and Kolter’19], as our bound in Theorem 3.1 has (tighter) dependencies on the same data-dependent quantities as theirs. We would address this concern by using the same argument as these prior works --- for networks trained in practice, the dependencies of these bounds on depth will be mild, as the quantities in these bounds will be well-behaved. For example, [Arora et. al’18, Nagarajan and Kolter’19] empirically observe networks trained in practice to have good Lipschitz constants and stability on the training sample.
>
> In fact, *in the worst case*, it would appear that some sort of exponential depth dependency is unavoidable, as Bartlett et. al’17 and Golowich et. al’17 lower bound the worst-case complexity of neural networks with products of the weight matrix norms. The goal of our work (and prior work [Wei and Ma’19, Nagarajan and Kolter’19]) is to identify properties which result in good generalization of neural networks *in practice*, not in the worst case. Our theory and experiments identify the all-layer margin as such a property.
>
> Responses to specific points below:
>
> --- “claims ... that the analysis removes exponential dependency of capacity on depth ... don’t think this is possible in the worst case.”
> --- “The paper need to make it clear that the bound only avoids explicit exponential dependence on depth.”
> --- “Note that the bound could still be interesting if this quantity is small for real world networks of interest. However the paper currently lacks this discussion.”
> --- “The paper provides a lower bound for this quantity in terms of layer norms, network Jacobian norms, and output margin - which can be large.”
>
> As noted above, avoiding an *explicit* exponential dependency is likely the best we can hope for, since lower bounds imply that complexity will be exponential in depth in the worst case. For networks trained in practice, our all-layer margin bound will be much better than the worst case. In Section 3, we analytically lower bound the all-layer margin in terms of the hidden layer and Jacobian norms computed on the training data, which were demonstrated to be well-behaved in practice by [Arora et. al’18, Nagarajan and Kolter’19].
>
> In our revision, we have added more qualifiers emphasizing that the bound will be small if the Jacobian and hidden layer norms on the training data are small (as is the case in practice). We also note in the revision that the Jacobian norms, hidden layer norms, and output margin will be well-behaved in practice in the first paragraph of Section 3.
>
> --- “robust generalization bound, which is nice ... depends on a hard to compute quantity - \kapp^{adv}, making it less clear about the utility of such a bound.”
>
> The quantity \kappa^{adv} depends only on the Jacobian and hidden layer norms and output margin in the adversarial perturbation neighborhood, and our bound suggests that regularizing these quantities during training could improve the robust generalization. In Section 5 Table 2 of the revision, we present preliminary experiments confirming this claim: our AMO algorithm improves upon the baseline method [Madry et. al ‘17] by around 6% in robust classification accuracy.
>
> --- “I am curious if authors experiments with dropout, as it also adds “perturbations” to the output activations of each layer, and if the gains are orthogonal to dropout.”
>
> We experimented with using dropout for the same WideResNet architectures and obtained ~0.2% accuracy improvement after tuning. Thus, our AMO algorithm outperforms dropout by at least ~0.5%, indicating that the adversarial (as opposed to random, for dropout) nature of the AMO perturbation results in better generalization. We also found that AMO + dropout performed worse than our models trained with only AMO, though it still improved over models trained without AMO. We suspect that combining these two regularizers inhibits training too much.
>
> --- “Also how does the performance of AMO compared with adversarial training methods (PGD) with small \epsilon?”
>
> For the clean classification setting, we are not aware of any work demonstrating that adversarial training methods can improve clean test accuracy. In fact, it has been observed that robust training can often hurt clean classification [Tsipras et. al’19, Zhang et. al’19, Raghunathan et. al’19].
>
> For robust classification, in Section 5, Table 2 of the revision, we provide preliminary results showing that AMO gives around 6% percent robust accuracy gains over adversarial training baselines (PGD, [Madry et. al’17]). For the final version of the paper, we will compare against additional baselines and attack models in the adversarially robust setting.

---

### Official Review · AnonReviewer2 · 2019-10-23
**Official Blind Review #2**

**Rating:** 8

**Review:**

The paper proposes a new generalization bound for deep neural networks and develops a regularizer which optimize quantities related to the bound and improve generalization error on competitive baselines. The paper treats deep neural network as a composition of functions (i.e. the layers) and introduces a new complexity measure, all layer margins, which depends on norm-based perturbation simultaneously applied to all the hidden layers of the neural networks. The all-layer margin can be seen as a generalization of the popular margin definition in SVM’s. Similarly, a generalization bound can be derived based on such all layer margins. The paper also shows that the bound has nice properties that improve upon some short-comings of previous efforts and compare favorably in terms of tightness. The paper also shows that this bound can be adapted to adversarial robustness of the deep model. Finally, a practical algorithm is proposed to optimize the approximation of the all-layer margin and improve upon the baseline models (i.e. trained with only cross-entropy) in a non-trivial manner.

I really like the paper! Jiang et al. 19 showed a surprisingly strong correlation between the margins at the hidden layers and generalization, which begs the question whether a generalization bound can be proven based on these perturbations. This paper shows that it is indeed the case with a much more sophisticated yet intuitive definition of margin that depends all hidden layers at the same time. Other than the nice theoretical properties outlined in the paper, the most exciting property of the all layer margin is that it can be directly optimized which the paper shows at the end. On the technical front, I have carefully read section 2 and Appendix A and read the other proofs at a coarser granularity. Barring some small notational errors, I did not find major errors in the derivation since standard techniques are used after the definition of the all layer margin. For the algorithm, while I have some questions about the thoroughness and accuracy of the approximation the results are quite compelling.

I think this paper should be accepted as I consider generalization to be one of the most important topics in deep learning and the paper is a valuable contribution both for theory and practices. In addition, the paper is relatively easy to follow for a topic so technically involved.

Barring my endorsement, I have some small complaints and questions:
    1. The notation for summation in the definition of \psi is quite difficult to parse. It would be good to either adopt a more explicit notation or have some extra clarification.

    2. The symbol for function F and the symbol for function class \mathcal{F} are sometimes mixed. For example, in lemma 2.1, the m_F on LHS should be m \circ \mathcal{F}.

    3. I am a little confused on the role of section 3 since, like the paper states, the bound in section 3 is always looser than section 2. Is this bound just for comparison with previous work or provide a tractable upper bound? If so this should be discussed.

    4. In lemma B.1, why are there 2 reference matrices and how should they be chosen?

    5. Condition A.1 assumes the function classes of each layer have some kind of bounded complexity. In a feedforward neural network, a single \mathcal{F}_i is just a linear operator which has pretty small complexity, but how about networks with residual connections? Is this assumption still reasonable? Likewise, since the proposed AMO is applied to residual networks, I want to hear the authors thought on this.

    6. Following the previous point, perhaps it would be nice to have some experiments on conventional feedforward networks and applied the proposed algorithm to all layers instead of skipping the contents of the residual block.

    7. Can the bound be accurately approximated? To me this seems almost combinatorially hard. If so, do the authors expect it to be non-vacuous? Given the results of Jiang et al., it seems that it is plausible to obtain fairly tight bound conditioned on the data and architecture.

    8. Adversarial robustness is a highly active and empirical-results-driven field, so some experiments on the defense against adversarial attack would greatly strengthen the theoretical results. Otherwise it’s hard to gauge how useful it is.


**Experience Assessment:**

I have published one or two papers in this area.

**Review Assessment: Checking Correctness Of Derivations And Theory:**

I assessed the sensibility of the derivations and theory.

**Review Assessment: Checking Correctness Of Experiments:**

I carefully checked the experiments.

**Review Assessment: Thoroughness In Paper Reading:**

I read the paper at least twice and used my best judgement in assessing the paper.

---

> ### Author Response · Authors · 2019-11-15
> **Response**
>
> We thank the reviewer for the comments and feedback. The review noted that our paper “is a valuable contribution both for theory and practices” and “relatively easy to follow for a topic so technically involved.” We will incorporate feedback and comments into the final version of our paper.
>
> Responses to specific points below:
>
> --- “I am a little confused on the role of section 3 … Is this bound just for comparison with previous work or provide a tractable upper bound?”
>
> The purpose of Section 3 is to show that the all-layer margin can be used to derive tighter generalization bounds depending on *known and computable* quantities --- our revision clarifies this point in the first paragraph of Section 3. Although the all-layer margin is likely hard to compute, Theorem 3.1 demonstrates that we can bound it by a function of the Jacobian norms, hidden layer norms, and output margin. This also lets us demonstrate that our bound improves upon prior work [Nagarajan and Kolter’19, Wei and Ma’19], which had worse dependencies on these same quantities. This also suggests that large all-layer margin intuitively encourages better Lipschitz-ness of the model.
>
> --- “In lemma B.1, why are there 2 reference matrices and how should they be chosen?”
>
> Our bound measures the complexity of the weight matrices in two norms, Frobenius norm and (1, 1) norm, so the use of two reference matrices just allows additional flexibility in computing the bound. In most of the cases, A_{(i)} and B_{(i)} can be chosen to be 0. If one has a prior belief before training that the weights will be close to some matrix A_{(i)} in Frobenius norm or a different matrix B_{(i)} in (1,1)-norm, then we would set the reference matrices to those values. (E.g., in certain cases, one may have the prior belief that the weight matrix is close to the identity matrix.)
>
> --- “In a feedforward neural network, a single \mathcal{F}_i is just a linear operator which has pretty small complexity, but how about networks with residual connections? ... since the proposed AMO is applied to residual networks, I want to hear the authors thought on this.”
>
> In standard resnets, the residual connections will not add any complexity since they are fixed functions, so the complexities of the individual layers will remain the same as in the feedforward case.
>
> It is possible to extend the all-layer margin beyond feedforward networks to networks with arbitrary computational graphs. For ResNet, the resulting bound would look like the sum of the complexities of the weight matrices normalized by this generalized all-layer margin, which matches the form of the bound for the feedforward case. The simplest way to perform this extension is to have a perturbation \delta at every node in the computational graph which is scaled by the norm of all its inputs.
>
> --- “perhaps it would be nice to have some experiments on conventional feedforward networks”
>
> For the VGG-19 architecture on CIFAR10, our AMO algorithm improves over standard SGD by around 0.5% accuracy. (See Table 3 in Section F.1 of the revision.)
>
> --- “applied the proposed algorithm to all layers instead of skipping the contents of the residual block.”
>
> Our AMO experiments with WideResNet do in fact apply the perturbations after all the convolution layers and don’t skip the contents of the residual block.
>
> --- “Can the bound be accurately approximated? … If so, do the authors expect it to be non-vacuous?”
>
> It does not appear that the all-layer margin can be accurately approximated —- doing so would require solving a seemingly intractable optimization problem. It also seems unlikely that the bound would be non-vacuous, since there are no known non-vacuous bounds for the  architectures and training algorithms used in practice (such as ResNets trained with vanilla SGD). Existing non-vacuous bounds appear to require specialized procedures both for obtaining the network and tuning the generalization bound [Dziugaite and Roy’17, Zhou et. al’19].
>
> Our view is also that a generalization bound does not need to be non-vacuous for it to be meaningful or informative, as the bound could still reveal the important quantities for generalization, which informs the design of regularizers. We believe this is an advantage of our all-layer margin --- the generalization bound is simple and easy to interpret, and we use it to design a regularizer which performs well in practice.
>
> --- “some experiments on the defense against adversarial attack would greatly strengthen the theoretical results”
>
> In Section 5, Table 2 of the revision, we have added preliminary results in the robust classification setting demonstrating improved performance over the robust training algorithm of [Madry et. al ‘17]. Our adversarially robust AMO procedure improves the robust accuracy by around 6% on PGD attacks. These results are based on preliminary experiments; for the final version of the paper, we will compare against additional baselines and attack models.

---

> > ### Comment · AnonReviewer2 · 2019-11-15
> > **Thank you for the response.**
> >
> > I thank the authors for the detailed response and new experiments despite that fact that I already recommended acceptance. All of my questions have been properly addressed.
> > It is interesting to see that the regularizer did not improve VGG-19 by a significant amount. It can be worthwhile to investigate why in the future work.

---

> > > ### Author Response · Authors · 2019-11-15
> > > **VGG-19 Improvement**
> > >
> > > Thank you for reading our response. It is indeed interesting that AMO doesn’t improve VGG as much as it does WideResNet models. We suspect that this is partially due to the fact that WideResNet models have larger capacity, but we agree that this is a question worthy of future investigation. However, we will note that the improvement is still there --- our algorithm improves VGG-19 by around ~0.6% for CIFAR10, which is only ~0.1% less than the improvement for our WideResNet-16-10 model and ~0.2% less than the improvement for our WideResNet-28-10 model for CIFAR10.

---

### Official Review · AnonReviewer4 · 2019-10-24
**Official Blind Review #4**

**Rating:** 8

**Review:**

The paper presents a novel and interesting way to measure the margin in the context of deep networks that removes the exponential dependency of depth in the corresponding generalization bounds. The key idea is to stabilize not only with respect to the input perturbations (which most existing works do and end up obtaining an exponential dependency), but simultaneous perturbations at every layer. It also shown that these ideas readily can be extended to provide bounds in the adversarial classification case. Finally, preliminary positive results on benchmarks showing how such an all-layer margin can be maximized during training are presented.

Major comments:
1. I really like the fresh idea of simultaneous perturbations based analysis. Though it is clear that the proposed all-layer margin is upper bounded by margins with single perturbation, it is indeed non-trivial and insightful to understand that the same alleviates the exponential dependency of depth. More specifically, I think claim 2.1 is the the most insightful result, though simple to prove in hindsight. It would greatly help the readers if simple figures are used to explain this insightful result in the final manuscript.

2. I think overall the paper was a pleasure to read especially with the way simplified analysis is presented in the main paper while postponing details to the Appendix.  It will be great if theorem 3.1 can further be simplified in notation and details just to present the main result that linear dependency is achieved via the lower bound for all-layer margin and to show improvement over existing bounds.

Minor comments:
1. Though results in sec5 are encouraging it would have been nice if it is shown that narrow and deep architectures benefit from such a regularization. As per the authors of WRN, the optimal architectures are not very deep.

2. Though linear dependency is a useful step, it does not still reflect the observation in practice that in many applications deep networks generalize better than the shallow ones. Any comments towards this might help.

3. There seem to be some typos: second min in (2.2) etc.

**Experience Assessment:**

I have read many papers in this area.

**Review Assessment: Checking Correctness Of Derivations And Theory:**

I assessed the sensibility of the derivations and theory.

**Review Assessment: Checking Correctness Of Experiments:**

I assessed the sensibility of the experiments.

**Review Assessment: Thoroughness In Paper Reading:**

I read the paper at least twice and used my best judgement in assessing the paper.

---

> ### Author Response · Authors · 2019-11-15
> **Response**
>
> We thank the reviewer for the comments. The review noted that “overall the paper was a pleasure to read.” Though our proof techniques are simpler than prior work, our all-layer margin “alleviates the exponential dependency of depth” in a “non-trivial and insightful” way. We will incorporate feedback and comments into the final version of our paper.
>
> Responses to specific points below:
>
> --- “Though results in sec5 are encouraging it would have been nice if it is shown that narrow and deep architectures benefit from such a regularization.”
>
> For a deeper but narrow architecture such as ResNet-52, we found that AMO improves performance when there is no data augmentation. However, with data augmentation, AMO gives less relative performance boost for ResNet-52 than for WideResNet architectures. We suspect that this is because the WideResNet models we use have a larger capacity than ResNet-52. Larger capacity allows the model to achieve a smaller perturbed loss (and therefore larger all-layer margin). Empirically, we observe that larger models do indeed obtain smaller perturbed loss.
>
> We note that the importance of capacity has also been observed for adversarial robustness, which also involves a min-max perturbed loss objective (see Section 4 of [Madry et. al’17]).
>
> --- “Though linear dependency is a useful step, it does not still reflect the observation in practice that in many applications deep networks generalize better than the shallow ones.”
>
> We agree that understanding the role of depth in generalization is an important theoretical question. Though this question largely remains open, we note that our bound can possibly improve with the depth of the network. At a high level, adding depth increases the expressivity of the network --- adding an extra layer could allow the model to fit the data with larger all-layer margin and *less* complexity required at each individual layer, making the overall bound better. To phrase this more technically: a deeper and more expressive model could fit the training data with smaller weight matrix, Jacobian, and hidden layer norms and larger output margin. This could imply a better bound for Theorem 3.1.

---

### Official Review · AnonReviewer3 · 2019-10-27
**Official Blind Review #3**

**Rating:** 6

**Review:**

The paper presents a bound on the generalization error of a deep network in terms of margin at each layer of the network.  The starting premise is that extending the existing margin generalization bounds to deep networks worsen exponentially with the depth of the
network. Recent work which removed that exponential dependency is
claimed to require a more involved proof and complicated dependence on
input.  The paper provides a new bound that is simpler
and tighter.

A second contribution is to extend their bounds to robust classifier.
Since their bounds depend on instance-specific margins, the extension
to the robust case is straightforward. They just need to relax the
margin to the robustness boundary of the input.

Finally, they present a new algorithm motivated by their bounds, that
maximized margin on all layers.  They show that the resultant network has much lower error than standard training.

The paper is well-presented and in spite of being theoretical is very nicely developed so that the main contributions come out clearly to non-specialists too.

A few minor comments:
The inner min in Equation 2.2 seems to be a typo.

In Theorem 2.1, there is typo around the definition of \xi.
Below thoerem 2.1, the phrase "depend on the q-th moment" has 'q' undefined.

Typo "is has a" in Theorem 3.1


**Experience Assessment:**

I do not know much about this area.

**Review Assessment: Checking Correctness Of Derivations And Theory:**

I did not assess the derivations or theory.

**Review Assessment: Checking Correctness Of Experiments:**

I did not assess the experiments.

**Review Assessment: Thoroughness In Paper Reading:**

I read the paper at least twice and used my best judgement in assessing the paper.

---

> ### Author Response · Authors · 2019-11-15
> **Response**
>
> We thank the reviewer for the comments.  The review noted that our work “provides a new bound that is simpler and tighter” than existing output margin-based generalization bounds, and our results are presented so that they “come out clearly to non-specialists too”. We also incorporated the comments on the typos into our revision.

---

### Author Response · Authors · 2019-11-15
**Summary of Revisions**

Besides incorporating the specific feedback of the reviewers, the main change we made in the revision was demonstrating that AMO (our proposed regularization algorithm) can improve performance for robust classification. We combine AMO and the robust training algorithm of [Madry et al.’17] in the robust classification setting. In Section 5, Table 2, we present preliminary experimental results on CIFAR10 demonstrating that our robust AMO algorithm achieves ~6% improvement in robustness on a PGD attack model over the baseline algorithm of [Madry et. al’17]. This validates our theoretical results in Section 4 and suggests that AMO is a promising procedure for improving adversarial robustness.

---

### Decision · Program_Chairs · 2019-12-19

**Decision:**

Accept (Poster)

**Comment:**

This works presents a new and interesting notion of margin for deep neural networks (that incorporates representation at all layers). It then develops generalization bounds based on the introduced margin. The reviewers pointed some concerns, including some notation issues, complexity in case of residual networks, removal of exponential dependence on depth,  and dependence on a hard to compute quantity - \kapp^{adv}. Some of these concerns were addressed by the authors. At the end, most of the reviewers find the notion of all-layer margin introduced in this paper a very novel and promising idea for characterizing generalization in deep networks. Agreeing with reviewers, I recommend accept. However, I request the authors to accommodate remaining comments /concerns raised by R1 in the final version of your paper. In particular, in your response to R1 you mentioned for one case you saw improvement even with dropout, but that is not mentioned in the revision; Please include related details in the draft.